# Using global reanalysis data to quantify and correct airflow distortion bias in shipborne wind speed measurements

Sebastian Landwehr[1], Iris Thurnherr[2], Nicolas Cassar[4], Martin Gysel-Beer[1], and Julia Schmale[1]

[1]Paul Scherrer Institute, Laboratory of Atmospheric Chemistry, Villigen, Switzerland
[2]ETH, Institute for Atmospheric and Climate Science, Zurich, Switzerland
[4]Division of Earth and Ocean Sciences, Nicholas School of the Environment, Duke University, Durham, USA

**Correspondence:** Sebastian Landwehr (sebastian.landwehr@psi.ch) and Julia Schmale (julia.schmale@psi.ch)

**Abstract.** At sea, wind forcing is responsible for the formation and development of surface waves and represents an important source of near surface turbulence. Therefore, processes related to near surface turbulence and wave breaking, such as sea spray emission and air-sea gas exchange are often parametrised with wind speed. Shipborne wind speed measurements thus provide highly relevant observations. They can, however, be compromised by flow distortion due to the ship's structure and objects nearby the anemometer that modify the airflow, leading to a deflection of the apparent wind direction and positive or negative acceleration of the apparent wind speed. The resulting errors in the estimated true wind speed can be greatly magnified at low wind speeds. For some research ships, correction factors have been derived from computational fluid dynamic models or through direct comparison with wind speed measurements from buoys. These correction factors can, however, lose their validity due to changes of the structures nearby the anemometer and thus require frequent re-evaluation, which is costly in either computational power or ship time. Here we evaluate if global weather forecast model data can be used to quantify the flow distortion bias in shipborne wind speed measurements. The method is tested on data from the Antarctic Circumnavigation Expedition (ACE) on board the R/V *Akademik Tryoshnikov*, which are compared with ERA-5 reanalysis wind speeds. We find that, depending on the relative wind direction, the relative wind speed and direction measurements are biased by $-36\%$ to $+21\%$ and $-17°$ to $+12°$, respectively. The resulting error in the true wind speed is $+11\%$ on average but ranges from $-4\%$ to $+41\%$ (5th and 95th percentile). After applying the bias correction, the uncertainty in the true wind speed is reduced to $\pm5\%$ and depends mainly on the average accuracy of the ERA-5 data over the period of the experiment. The obvious drawback of this approach is the potential intrusion of model bias in the correction factors. We show that this problem can be somewhat mediated when the error propagation in the true wind correction is accounted for and used to weight the observations. We discuss the potential caveats and limitations of this approach and conclude that it can be used to quantify flow distortion bias for ships that operate on a global scale. The method can also be valuable to verify Computational Fluid Dynamic studies of airflow distortion on research vessels.

# 1    Introduction

Wind speed is an important factor for air-sea interaction. With increasing wind speed small instabilities at the air-water interface grow to waves that modify both, the surface roughness and the airflow near the surface. Wave breaking leads to localised generation of turbulence, entrainment of air and production of sea spray. As these wind-driven processes control the exchange of momentum, heat, trace gases and particles between the atmosphere and the ocean, wind speed is often used to parametrise air-sea exchange processes. For example, gas transfer is typically parametrised solely by wind speed with the proposed dependencies ranging from nearly linear, (e.g. Krall and Jähne, 2019), over quadratic (e.g. Ho et al., 2006) to cubic (Wanninkhof and McGillis, 1999). For the production of sea spray, most parametrisations are based on Monahan et al. (1986), who suggested that the sea spray flux could be modelled as function of wind speed with a power law exponent of 3.41. This strong dependency on wind speed means that a relatively low uncertainty in the wind speed translates to potentially significant uncertainties in the parametrised exchange processes. In case of sea spray production being parametrised with wind speed to the power of 3.41, a 10% error in the wind speed results in an error in the predicted sea spray production of 38%.

Within the turbulent surface layer that extends from a few millimetres to a few tens of meters above the sea surface, the wind speed changes with height, whereby the shape of the wind speed profile depends on the atmospheric stability (Monin and Obukhov, 1954). In order to make observations comparable, the wind speed is typically reported as equivalent to the wind speed 10 meters above sea level and neutral stability ($u_{10N}$):

$$u_{10N} = u(z) - \frac{u_*}{\kappa}\left[\log\left(\frac{z}{10}\right) - \Psi_u\left(\frac{z}{L_*}\right)\right],\tag{1}$$

where $u_*$ is the friction velocity, which is related to $u_{10N}$ via the surface drag coefficient ($C_{D10N} = u_*^2\, u_{10N}^{-2}$) (Smith, 1988; Fairall et al., 2003), $\kappa = 0.4$ is the van Karman constant, and $\Psi_u\left(\frac{z}{L_*}\right)$ is a dimensionless function of the measurement height ($z$) and the Obukhov length scale ($L_*$) that accounts for the effects of atmospheric stability that lead to a deviation from the logarithmic profile. Obukhov length scale characterizes the relative contributions to turbulent kinetic energy from buoyant production and shear production and is given by:

$$L_* = \frac{u_*^3}{\kappa\, g(\langle w'T'\rangle/T_v + 0.61\langle w'q'\rangle)},\tag{2}$$

where $g$ denotes the acceleration due to gravity, $T_v$ the surface virtual temperature, and $\langle w'T'\rangle$ and $\langle w'q'\rangle$ are the turbulent surface sensible and latent heat fluxes, respectively.

Aboard ships and on buoys, wind speed measurements are typically performed with 2-D or 3-D anemometers mounted on exposed locations. The measurement platform and smaller structures near the anemometer can cause a distortion of the airflow and, thereby, reduce the accuracy of the in situ wind speed and direction measurements. Moat et al. (2005) report a typical range of wind speed bias of +11% to -100% for anemometer locations above the bridge of research and cargo vessels. For buoys, the ratio of the sensor's height above the main structure to the dimension of the structure is much higher, so that airflow

distortion is typically lower, in the order of 5% to 10% (e.g. Emond et al., 2012; Bigorre et al., 2012). Therefore, wind speed measurements from buoys are generally taken as reference for the evaluation of other wind speed products.

Remote sensing systems (altimeter, scatterometer, and microwave radiometer) offer global observations of surface wind speed related quantities from space. The observed signals are calibrated and validated against surface wind speed observations from buoy networks (Young et al., 2017; Stopa and Cheung, 2014; Jones et al., 2016; Schmidt et al., 2017; Zhang et al., 2018) and from voluntarily observing ships (e.g. Bourassa et al., 2003).

Global atmospheric weather forecast and reanalysis products are provided, for example, by the European Center for Medium-range Weather Forecasts (ECMWF) and the United States' National Centers of Environmental Prediction (NCEP). Over the ocean, the accuracy of the numerical models is improved by the assimilation of in situ observations from ships, buoys, and satellite derived wind speeds. Several studies have investigated the performance of numerical weather models via comparison to in situ observations from ships, buoys, and wave-gliders (e.g. Li et al., 2013; Stopa and Cheung, 2014; Jones et al., 2016; Schmidt et al., 2017; Zhang et al., 2018; Belmonte Rivas and Stoffelen, 2019). They document a significant improvement in the precision and accuracy of these models over the last decades, but also variable model bias that can depend on region and season.

For research experiments in the open ocean, especially in remote areas such as the Southern Ocean, where only few observations are available, it is desirable to use the shipborne wind speed measurements that offer a higher temporal resolution than remote sensing and numerical weather model products. Flow distortion can, however, lead to biased wind speed and direction estimates that affect the comparison of wind speed-related observations, if they have been made from different ships, but also if they have been made from the same ship but at a different relative wind direction. Corrections for airflow distortion have been derived from Computational Fluid Dynamics (CFD) models (e.g Popinet et al., 2004; O'Sullivan et al., 2013). This approach requires a detailed 3D model of the ship's structure. Due to computational limitations, such CFD simulations are often performed for a limited number of relative wind directions, and small ship structures cannot be resolved (Moat et al., 2005; Popinet et al., 2004). However, small structures in the vicinity of the measurement site can have significant impact on the pattern of the airflow (O'Sullivan et al., 2013). Furthermore, modifications to the surrounding structures may invalidate the results obtained by prior CFD studies. The bias in wind speed and wind direction is mainly dependent on the location of the wind sensor and the relative wind direction (Popinet et al., 2004). In the results of their CFD simulation, O'Sullivan et al. (2013) observed changes in the relative wind speed bias in dependence of the pitch and roll of the ship as well as the magnitude of the relative wind speed. An experimental verification of these findings is, however, outstanding. Landwehr et al. (2015) quantified flow distortion on a research vessel via direct comparison to wind speed measurements from a nearby buoy. Due to the sparsity of the buoy networks this approach is, however, not feasible for most experiments, since it would require the dedication of ship time to visit one of these buoys.

A less direct approach would be the validation of shipborne wind measurements against calibrated remote sensing wind speeds. However, despite the growing number of wind-sensing satellites in the orbit, the frequency of overpasses at a single location is still small as altimeter sensors return to a location within 5 to 20 days and radiometer missions approximately twice per day (Young and Donelan, 2018).

In this work we explore the possibility of using numerical weather reanalysis products, which are constrained via the assimilation of ship, buoy and remote sensing wind speeds but fill the gaps between the observations with predictions based on state of the art process models. We develop a framework to detect and quantify flow distortion in shipborne measurements using reanalysis data from ERA-5 and apply it to wind speed measurements from the Antarctic Circumnavigation Expedition (ACE) in the Southern and Atlantic Ocean. Furthermore, we will discuss possible concerns of this method such as the effect of the low temporal and spatial resolution and model biases of the reanalysis products.

Sect. 2 provides a short overview of studies that have evaluated ERA-5 and its predecessor ERA-Interim. The data used in this study and the methodology are described in Sect. 3. The results are presented and discussed in Sect. 4 and Sect. 5, and conclusions are drawn in Sect. 6.

## 2 Performance of ERA-Interim and ERA-5

The performance of ERA-5 and its predecessor ERA-Interim has been evaluated in several studies: Stopa and Cheung (2014) and (Zhang et al., 2018) used wind speeds from buoys to validate wind speed (and wave height) from remote sensing products and ERA-Interim reanalysis data. Both studies found that the bias in the ERA-Interim wind speeds varied between regions and for different seasons. For the latitude band $60°$S to $40°$S and the period December-January-February both studies report that ERA-Interim wind speeds are biased high compared to the satellite wind speeds. During ACE, the majority of data was collected in this latitude band. ERA-Interim wind speeds are also reported to be biased low in the equatorial region west of Africa which the ship crosses during leg 4 (see for example Fig. 4 (a) in (Young and Donelan, 2018)). The studies also report seasonal and regional differences in the agreement between buoy and satellite wind speeds. Since buoy measurements are performed at heights around 1 meter above sea level, these were converted to $u_{10}$ via Eq. (1), however neglecting the stability correction term.

A recent analysis (Young and Donelan, 2018), however, showed that some of the seasonal and regional bias between buoy and satellite wind speeds may be an artifact, caused by the neglecting of the stability correction in Eq. (1). They provide correction factors for radiometer and altimeter wind speeds, which account for the effects of stability on the wind speed profile as well as the wind speed dependence of the effective sensing height of the two systems.

Utilizing re-calibrated scatterometer wind speeds from ASCAT, Belmonte Rivas and Stoffelen (2019) characterized ERA-Interim and ERA-5 wind vectors for the year 2016. In comparison to ASCAT they reported mean zonal and meridional wind speed bias of less than $0.5\,\mathrm{m\,s^{-1}}$ for ERA-Interim and less than $0.3\,\mathrm{m\,s^{-1}}$ for ERA-5. They also report root mean square errors (RMSE) of less than $3\,\mathrm{m\,s^{-1}}$ and less than $2.5\,\mathrm{m\,s^{-1}}$, for ERA-Interim and ERA-5, respectively. Global maps of mean wind speed difference (ASCAT minus ERA-Interim and ASCAT minus ERA-5) presented by Belmonte Rivas and Stoffelen (2019) still show regional variability of the bias against ASCAT wind speeds but also a clear reduction for ERA-5, compared to ERA-Interim. Figure 5 in Belmonte Rivas and Stoffelen (2019) shows annual mean bias in ERA-5 zonal and meridional wind speeds against ASCAT of up to $1\,\mathrm{m\,s^{-1}}$ in the equatorial region west of Africa, and less than $0.5\,\mathrm{m\,s^{-1}}$ in the Southern Ocean.

It is worth mentioning that some studies have evaluated ERA-Interim in the Southern Ocean against in situ observations from ships (Li et al., 2013; Jones et al., 2016). These studies reported mean bias and RMSE of a few $\mathrm{m\,s^{-1}}$, which are, however, quite variable between the different ships. These results may be affected by flow distortion bias in the shipborne measurements. Li et al. (2013) reported that, where data was available from more than one wind sensor on a single ship, they used the consistency between those readings as measure to filter potentially affected data.

## 3  Data and Methods

### 3.1  Shipborne wind measurements during the Antarctic Circumnavigation Expedition

The Antarctic Circumnavigation Expedition (ACE) was conducted aboard the R/V-*Akademik Tryoshnikov*. A total of 22 international projects where involved and measured a wide range of variables in the atmosphere, the ocean, on Subantarctic islands and the Antarctic continent (Walton and Thomas, 2018). The ship moved from Bremerhaven, Germany to Cape Town, South Africa (leg 0), from Cape Town through the Indian Ocean to Hobart, Australia (leg 1); from Hobart via the Pacific Ocean to Punta Arenas, Chile (leg 2); from Punta Arenas through the Atlantic Ocean back to Cape Town (leg 3); and finally north to Bremerhaven (leg 4) (Schmale et al., 2019a). Therefore, the dataset covers a full circumnavigation of the Antarctic continent between $34°$S and $78°$S and two Atlantic transects from $34°$S to $53°$N. Figure 1 shows a map of the cruise track CITEDATASET.

In situ wind speed was measured aboard R/V *Akademik Tryoshnikov* with two 2-D sonic anemometers (model: WS425 and WMT702), which were operated as part of an automated weather station (AWS; model: AWS420, Vaisala). The anemometers were mounted on 2 m long vertical poles, which are attached to the two topmost side arms on the startboard side and portside of the main mast (see Fig. 2). This places the measurement volumes, $\sim 3\,\mathrm{m}$ above the highest floor of the main mast, $\sim 8\,\mathrm{m}$ above the monkey island (the area above the bridge), and $\sim 31.5$ meters above mean sea level (m a.s.l). In the following, measurements from the anemometer located on the starboard side are labelled with the suffix ("stbd"), measurements on the port side with the suffix ("port").

Figure 3 shows photographs of the set-up that where taken on a visit to the ship before the cruise. At the level of the anemometers, the main mast and the radar antenna on the starboard side represent two obstacles to the airflow, while the radar antenna on the port side is located more than 1.5 m below the level of anemometers.

The recording and preprocessing of the AWS data is documented in (Pina Estany and Thomas, 2019). The AWS provides a record of the measured relative wind speed ($S$) and relative wind direction ($D$) as well as the ship's heading ($\Phi_H$) with a 3 second resolution. The convention used here is $D = 0°$ if the ship is pointed into the wind and $D = 90°$ for wind coming from starboard. Where $D$ is used as x-axis in the figures, we have values reaching from $-180°$ to $+180°$ in order to create a panorama. Negative values of $D$ denote wind from the port side and positive values from the starboard side, respectively.

True wind speed and direction were also provided by the AWS. However, for the internal correction the AWS was programmed to assume that the ship's course ($\Phi_C$) would always equal the heading (communications with Vaisala customer support). This assumption neglects situations when the ship's velocity is not aligned with the ship's heading, e.g. when the

ship is slowly drifting sideways during a station, and results in an underestimation of the true wind speed (Smith et al., 1999). For this data set, the difference between course and heading was higher than $10°$ for about 22% of the time. Therefore, we calculated the true wind ($\boldsymbol{u}$) in post-processing:

$$\boldsymbol{u} = \mathbf{H}_{es}\boldsymbol{r} + \boldsymbol{v}, \tag{3}$$

where, $\boldsymbol{v}$ is the ship's velocity, $\boldsymbol{r}$ is the observed relative wind vector in the ship's frame of reference, and $\mathbf{H}_{es}$ is the transformation from the instantaneous ship's frame of reference to the East, North, upward coordinate-system.

Here, the ship's reference system is defined as follows: x-axis along the ship's main axis and positive towards the bow, y-axis along the beam and positive towards port, z-axis positive upward. The vertical component will be neglected in the following analysis as the time averages of the vertical wind speed and the ship's vertical velocity component are negligible for the used averaging periods of 5 min and 1 hour. For the same reason, instantaneous pitch and roll of the ship are neglected. The 5 min average roll varied between $-2°$ and $+2°$, while the pitch varied only between $-1.15°$ and $-1.90°$. The simplified transformation from ship to earth reference system is given by:

$$\mathbf{H}_{es} = \begin{pmatrix} \cos\left(90° - \Phi_H\right) & \sin\left(90° - \Phi_H\right) \\ -\sin\left(90° - \Phi_H\right) & \cos\left(90° - \Phi_H\right) \end{pmatrix} \tag{4}$$

The relative wind vector is calculated from $S$ and $D$ as follows:

$$\boldsymbol{r} = S \begin{pmatrix} \cos\left(180° - D\right) \\ \sin\left(180° - D\right) \end{pmatrix} \tag{5}$$

During the first part of leg 0 (before 2016-11-27 10:00:00), the navigation data is only available at low time resolution (less than two samples per minute). For these data the ships velocity was recovered from the records of the ship's heading and the true and relative wind speed and direction under the a priori assumption $\Phi_C = \Phi_H$, which holds approximatley for $|\boldsymbol{v}| > 2\,\mathrm{m\,s^{-1}}$. If the re-calculated velocity of the ship was above $2\,\mathrm{m\,s^{-1}}$ the data were used in this analysis. The quality controlled record of the meteorological observations, which were recorded by the AWS can be obtained from (Landwehr et al., 2019).

### 3.2 Modelled relative wind speed based on ERA-5 and estimation of the flow distortion bias

We derive model predicted relative wind speed and direction from the ECMWF weather model reanalysis data ERA-5 (Copernicus Climate Change Service (C3S) (2017), 2017) as follows: The ERA-5 dataset is provided on a $0.25° \times 0.25°$ spatial resolution and one-hourly temporal resolution. From this, the nearest value in space and time was interpolated onto the ship's track at five-minute resolution. For details on the choice of the time resolution, see Sect. 3.4. The ERA-5 $u_{10\mathrm{N}}$ is used to calculate $u_*$, using the wind speed dependent neutral drag coefficient $C_{\mathrm{D10N}}$ provided in the COARE 3.5 bulk flux model (Edson et al., 2013). We use the ERA-5 sensible and latent heat fluxes together with $u_*$ to derive $L_{*\mathrm{M}}$ via Eq. 2. Using ERA-5 heat fluxes instead of the in situ air-sea temperature gradient may introduce  errors in the profile adjustment. The ERA-5 SST estimates agree within $\pm 1°\mathrm{C}$ with calibrated underway measurements of the bulk sea water temperature at a depth of five meters,

which are provided in Haumann et al. (2020) for 96% of the data. The same holds for the majority (86%) of the surface air temperature observations, however localized events with differnces in $T_a$ up to $8°C$ between ERA-5 and the observations were observed in the vicinity of some of the islands (See Section E in the Appendix).

Using Eq. (1) we calculate the predicted wind speed at the measurement height $z = 31.5\,\mathrm{m}$ which will in the following be termed $u_\mathrm{M}$ with the suffix "M" for model-derived. Note that, as pointed out by Edson et al. (2013), the wind speed profile should be evaluated relative to the water surface. However, since the surface water currents were not measured during ACE, we evaluate against a fixed earth reference frame. This may introduce small errors in the height and stability correction.

    Rearranging Eq. (3), the relative wind vector predicted by the model can be calculated as

$$r_\mathrm{M} = \mathbf{H}_{es}^{-1}[u_\mathrm{M} - v]. \tag{6}$$

The expected relative wind speed ($S_\mathrm{M}$) and direction ($D_\mathrm{M}$) can be derived from $r_\mathrm{M}$ via Eq. (5) and compared to the measurements from the port and starboard anemometer.

    The relation between observed wind speed ($S_\mathrm{m}$) and direction ($D_\mathrm{m}$) and the model predicted values $S_\mathrm{M}$ and $D_\mathrm{M}$ provide means to quantify the flow distortion correction factors:

$$\alpha_\mathrm{S}(D_\mathrm{m}) = \left\langle S_\mathrm{m}\, S_\mathrm{M}^{-1} \right\rangle \tag{7}$$

and

$$\delta_\mathrm{D}(D_\mathrm{m}) = \left\langle D_\mathrm{m} - D_\mathrm{M} \right\rangle, \tag{8}$$

where the angular brackets denote the average of the observations over sufficiently small intervals of the measured relative wind direction ($D_\mathrm{m}$). Due to the complexity of the structures near the anemometer, the acceleration factor ($\alpha_\mathrm{S}$) and the horizontal
deflection ($\delta_\mathrm{D}$) vary with the angle of attack. Here we account only for the horizontal variations that are given by measured relative wind direction ($D_\mathrm{m}$). This approach neglects any potential effects of the pitch and roll on the flow pattern of the airflow, however, small variations of the wind speed ratio from the port and starboard anemometers may be attributed to changes in the mean roll angle (see Appendix A). Effects of the pitch were not observed, but cannot be fully ruled out due to the symmetry of the measurement setup. However, the mean pitch varied by less than $1°$.

We apply the correction to the three-second time series of $D_m$ and $S_m$ in order to calculate a corrected relative wind speed

$$r_\mathbf{c} = (\alpha_\mathrm{S}^{-1}\, S_\mathrm{m}) \begin{pmatrix} \cos\left(180° - D_\mathrm{m} + \delta_\mathrm{D}\right) \\ \sin\left(180° - D_\mathrm{m} + \delta_\mathrm{D}\right) \end{pmatrix}, \tag{9}$$

which is then used to re-compute the corrected true wind speed ($u_\mathrm{c}$) via Eq. 3. Note that the surface sensible heat flux is approximately linearly related to $u_*$ and for this reason $L_*$ is approximately proportional to $u_*^{-2}$. Therefore we use an adjusted ($L_{*c} = L_{*\mathrm{M}}\, u_\mathrm{c}^2\, u_\mathrm{M}^{-2}$) to derive $u_{10\mathrm{N}}$ from the corrected true wind speed $u_\mathrm{c}$ via Eq. (1).

### 3.3   Uncertainty estimation

Errors in the used reference wind speed and direction will propagate into the estimates of the expected relative wind speed and direction and consequently of $\alpha_S$ and $\delta_D$. Due to the vector addition, errors in the ERA-5 wind speed and direction are less

severe for the bias estimation, if the ship is heading against the wind. However, if the ship is moving in the same direction as the wind, the vector addition leads to an amplification of the relative error. This effect is enhanced when the ratio of the ship speed to the wind speed increases. A detailed description of the error propagation can be found in Appendix B.

In order to estimate the uncertainty in the expected relative wind speed and direction, we vary the ERA-5 wind speed by $\pm 20\%$ or $0.5\,\mathrm{m\,s^{-1}}$ (whichever is larger) and the ERA-5 wind directions by $\pm 10°$. These values are based on the comparison of the in situ wind speeds from ACE with the ERA-5 predictions. We use the largest absolute deviation of $S_\mathrm{M}$ and $D_\mathrm{M}$, resulting from these combinations to estimate $\Delta S_\mathrm{M}$ and $\Delta D_\mathrm{M}$.

The accuracy of the relative wind speed and direction readings are taken from the data sheets for WMT702 (WS425): $\Delta S_\mathrm{m} =$ 
$1\%$ ($\Delta S_\mathrm{m} = 2\%$) and $\Delta D_\mathrm{m} = 2°$ for both models. We neglect the uncertainties in the velocity and heading measurements since these are small in comparison.

The uncertainty in the flow distortion bias estimates are given by

$$\Delta(\alpha_S) = \sqrt{(\Delta S_\mathrm{m})^2 + (\Delta S_\mathrm{M} \frac{S_\mathrm{m}}{(S_\mathrm{M})^{-2}})^2} \tag{10}$$

and

$$\Delta(\delta_D) = \sqrt{(\Delta D_\mathrm{m})^2 + (\Delta D_\mathrm{M})^2} \tag{11}$$

We use the estimated uncertainties as weights for the calculation of weighted means as the best estimates of $\alpha_S(D_\mathrm{m})$ and $\delta_D(D_\mathrm{m})$.

## 3.4   Choice of wind direction sectors and time averages

In the following we use relative wind direction sectors to calculate average wind speed ratios and wind direction differences.
These sectors have been chosen as a compromise between directional resolution and sample size. For $-140° < D_\mathrm{m} <= +180°$ we use an interval width of $5°$ to optimally resolve the variation of $\alpha_S$ and $\delta_D$ with $D_\mathrm{m}$. For $-180° < D_\mathrm{m} < -140°$, however, the interval width had to be reduced to $20°$ due to the small number of observations in this sector.

Time averages are necessary to obtain meaningful values of wind speed and direction. However, depending on the ship's layout, the flow distortion bias can be very sensitive to small changes in the relative wind direction. Thus, for the experimental
bias determination the dataset needs to be restricted to time intervals over which the relative wind direction did not change significantly. This is fulfilled more easily if the time intervals are short. Here we choose an averaging time of five-minutes for the analysis, but results do not change significantly for longer averaging times up to one hour (See Section C in the Appendix).

The wind direction sectors are used to calculate weighted averages of the wind speed ratios and wind direction differences. We approximate the error of the mean via bootstrapping: In a first step all estimates of $\alpha_S$ and $\delta_D$ in a given wind direction
sector, which are derived from the same $0.25° \times 0.25° \times 1\,\mathrm{hour}$ ERA-5 grid cell are averaged and treated as one independent sample. The resulting population is resampled 100 times and the standard deviation of the resulting 100 weighted averages is used as an estimate of the standard error of the mean.

### 3.5 Selection of data for the estimation of $\alpha_S$ and $\delta_D$

The data set used in this study amounts to 37835 and 37925 five-minute average observations from the port and starboard sensor. Subinterval variability of relative wind direction, ship's heading and speed for each five-minute interval was evaluated using one minute average data. Only samples where the subinterval variations of $D$ and $\Phi_H$ were less than $15°$ and the subinterval variation of $|v|$ was less than $1\,\mathrm{m\,s^{-1}}$ were used for the analysis. About 27% of the observations failed these criteria.

During leg 2 when the ship was south of $60°\mathrm{S}$ the wind speed and direction observations were reported to the Global Telecomunication System (GTS) under the call sign UBXH3 and partly used for data assimilation in the Integrated Forecast System (IFS) ECMWF (2016). ERA-5 reanalysis output, which may have been effected by the assimilation of observations from the R/V *Akademik Tryoshnikov* , was identified and excluded from this analysis. These were 277 hours in total. See Appendix section D for further details.

For the estimation of $\alpha_S$ and $\delta_D$ we further limit our dataset to observations, which were made in ERA-5 open ocean grid cells (i.e., where the ERA-5 land-sea mask and the sea ice fraction are equal to zero) and for which the distance from the ship's location to the nearest coastline was larger than $50\,\mathrm{km}$. We also restrict the data to situations where $-1.5 < z\,L_*^{-1} < 0.25$, in order to limit the magnitude of the stability correction in Eq. (1) to less than 10% of $u_{10\mathrm{N}}$.

If the ship is moving in the same direction and with approximately the same speed as the airflow, the relative wind speed will approach zero and the relative wind direction cannot be defined. Therefore, intervals where either $S_\mathrm{m}$ or $S_\mathrm{M}$ are smaller than $2\,\mathrm{m\,s^{-1}}$ are not used for the estimation of $\alpha_S$ and $\delta_D$. About 35% of the observations (15209 and 15397 five-minute intervals for port and starboard, respectively) passed all criteria and are used for the estimation of the flow distortion bias.

Faulty data transmission or undocumented interference with the sensors (birds, rimming, or heavy rain) can lead to errors in $\boldsymbol{r}_\mathrm{m}$ that affect a small number of observations. Further, local weather events may not be resolved in the ECMWF model, leading to large differences between $\boldsymbol{u}_\mathrm{M}$ and the true wind speed. In consequence some estimates of $\alpha_S$ and $\delta_D$ will deviate largely from main distribution and reduce the accuracy of the estimated mean values. We use a standard method to identify these outliers based on the interquartile range (IQR), i.e, the difference between the $75^\mathrm{th}$ and $25^\mathrm{th}$ percentile: for each wind direction interval, values that lay more than $1.5 \times \mathrm{IQR}$ above the $75^\mathrm{th}$ percentile or more than $1.5 \times \mathrm{IQR}$ below the $25^\mathrm{th}$ percentile are treated as outliers and are excluded from the calculations. For the paired estimation of $\alpha_S$ and $\delta_D$, a data point was excluded when it failed the criterion for either of the two. This method is termed IQR-filter in the following. The IQR-filter removed about 7% of the observations that had passed the above quality criteria. In total 13353 and 13529 five-minute samples passed all quality control criteria for the part and starboard sensor, respectively. This amounts to about 35% of the original available data from both sensors.

Note that the above described filtering methods are only used to derive the subset of data which is suitable to estimate $\alpha_S(D_\mathrm{m})$ and $\delta_D(D_\mathrm{m})$. The flow distortion correction factors are later applied to the full data set of $\boldsymbol{r}_\mathrm{m}$ in order to derive the corrected wind speed.

## 4 Results

### 4.1 Intercomparison of the measurements of port and starboard anemometers

In Fig. 4, the five-minute averaged relative wind speed and direction recorded by the two anemometers are compared against each other. The ratio of the port and starboard relative wind speeds and the difference of the relative wind direction are shown as functions of the relative wind direction and the subinterval variability of the relative wind direction.

The measurements from the starboard and port sensors agree best for $D \approx 0°$, but differences in the relative wind speed of up to $40\%$ occur at $D \approx \pm 90°$, where the lee side readings are affected by shadowing due to the main mast. The smaller variation of $S_{\mathrm{m,stbd}} S_{\mathrm{m,port}}^{-1}$ and $D_{\mathrm{m,stbd}} - D_{\mathrm{m,port}}$ at $D \approx -40°$ is likely caused by the wake of the small mast, which is mounted on the starboard side (see Fig. 2). This mast affects only the starboard side sensor. The difference between the relative wind direction measurements ranges between $-9°$ and $+12°$. For the relative wind direction sector $-170° < D < -135°$ both wind speed ratio and wind direction difference show a larger variability than for other sectors and only few data points passed the wind direction variability criterion (see Sect. 3.5). A possible explanation for this are turbulences caused by the structure of and emissions from the exhaust stack, which is located in this direction.

### 4.2 Average deflection and acceleration estimates based on ERA-5 wind speeds

The $\boldsymbol{u}_{\mathrm{M}}$ derived from ERA-5 are used to estimate $S_{\mathrm{M}}$ and $D_{\mathrm{M}}$ as described in Sect. 3.2. In Fig. 5 the average values of $S_{\mathrm{m,stbd}} S_{\mathrm{M}}^{-1}$ and $D_{\mathrm{m,stbd}} - D_{\mathrm{M}}$ per wind direction bin are displayed as function of $D_{\mathrm{m,stbd}}$. The distribution of the individual observations of $S_{\mathrm{m,stbd}} S_{\mathrm{M}}^{-1}$ and $D_{\mathrm{m,stbd}} - D_{\mathrm{M}}$ are shown as heat maps (bi-dimensional histogram). In this figure we show the samples which passed all quality control measures, except the IQR-filter. Although the variability of the individual ratios and direction differences is high, there is a clear trend of the flow distortion bias with relative wind direction: The ratio $S_{\mathrm{m,stbd}} S_{\mathrm{M}}^{-1}$ peaks at 1.2 for $D_{\mathrm{m,stbd}} \approx 0°$ and is close to 1.05 for $D_{\mathrm{m,stbd}} \approx 180°$. Two minima are visible at $D_{\mathrm{m,stbd}} \approx -90°$ and $D_{\mathrm{m,stbd}} \approx -40°$, which can be attributed to shadowing of the starboard sensor by the main mast and the radar antenna. For this wind directions the relative wind speed is underestimated by $-36\%$ and $-14\%$ respectively. The difference $D_{\mathrm{m,stbd}} - D_{\mathrm{M}}$ amounts to $\approx -5°$ for bow on wind direction. For the wind ward side $D_{\mathrm{m,stbd}} > 0°$ the relative wind vector is increasingly deflected away from the center line until $D_{\mathrm{m,stbd}} \approx +90°$, where the bias in $D_{\mathrm{m,stbd}}$ starts to decrease again. More abrupt variations in $D_{\mathrm{m,stbd}} - D_{\mathrm{M}}$ are observed for wind direction from port, for which the starboard sensor is in the wake of the main mast and the radar antenna.

The average variation of $S_{\mathrm{m,port}} S_{\mathrm{M}}^{-1}$ is shown in Fig. 6. For the port sensor the overestimation of the relative wind speed is largest at $D_{\mathrm{m,port}} \approx -15°$, where it amounts to $+18\%$. The strongest underestimation ($-25\%$) occurs at $D_{\mathrm{m,port}} \approx +90°$, where the port sensor is in the wake of the main mast.

Several studies (e.g. Stopa and Cheung, 2014; Zhang et al., 2018) suggest that during the Southern summer ERA-Interim wind speeds are biased high by $\approx 5\%$ in the latitude band $40°$S to $60°$S. In order to investigate, if this affects the derived correction, the bin averaged ratios $S_{\mathrm{m}} S_{\mathrm{M}}^{-1}$ for the port sensor are plotted in Fig. 6 for ship positions within and outside of the latitude band $40°$S to $60°$S separately. Figure 7 shows the number of "unique" five-minute intervals per sector, where

"unique" means that for each relative wind direction sector multiple matches with the same latitude×longitude×time grid box are counted only once. The ratio $S_m \, S_M^{-1}$ tends to be higher for ship positions north of $40°$S than for observations from the latitude band $40°$S to $60°$S for most relative wind direction sectors. For $-15° < D_m < 15°$, where a large number of observations allow for robust averages, the estimated ratios are on average $\approx 4\% \pm 2\%$ lower in the latitude band $40°$S to $60°$S than north of these latitudes. For latitudes south of $60°$S the number of five-minute averages that are not compromised by the assimilation of the observations into the IFS is too low for a wind sector resolved analysis.

## 4.3 Effect of the correction on the estimated true wind speeds

Figure 8 shows the distribution of $u_m$ versus $u_c$ (the measured versus corrected true wind speed). The correction of the measured wind vector via Eq. 9 tends to reduce the true wind speed but the magnitude of the correction varies by more than $5\,\mathrm{m\,s^{-1}}$. The effect of the correction on the estimates of $\boldsymbol{u}$ (and consequently $u_{10N}$) depends on the relative wind direction as well as the ratio of wind speed and ship velocity. Figure 9 shows the distribution of $u_m \, u_c^{-1}$ for the starboard sensor as function of the relative wind direction, as well as the histogram integrated over all relative wind directions. For 36% of the data the change in the true wind speed estimates by the correction is less than 5%. These observations, which are nearly unaffected by flow distortion, are almost exclusively from $|D_m| > 30°$. The effect of the correction on $u$ is strongest in magnitude for $-30° < D_m < +30°$, where the vector addition of true wind speed and ship velocity can lead to situations where $S \, |u|^{-1} \gg 1$. In most cases the correction leads to a lower estimate of $u$. The few cases in this sector where the correction leads to an increase of $u$ are related to situations when the ship is heading in the same direction into which the wind blows and has a higher speed than the wind.

For 45% and 27% of the data the effect of the correction on $u$ is stronger than 10% and 20%, respectively, while a bias stronger than $\pm 40\%$ occurs for 10% of the measurements. The percentiles of the distributions of $u_m \, u_c^{-1}$ for the port and the starboard sensor as well as for averages of the port and starboard wind speeds are summarised in Tab. 1.

Figure 10 shows the histograms of $u_{m,stbd} \, u_{m,port}^{-1}$ and $u_{c,stbd} \, u_{c,port}^{-1}$. The correction, which has been derived independently for each sensor, improves the agreement between the port and starboard wind speeds, as can be seen by the narrower distribution of the wind speed ratio.

## 4.4 Remaining uncertainty in the wind speed measurements

During ACE the correction for the measurement height ($\frac{u_*}{\kappa} \log\left(\frac{z}{10}\right)$) ranges from 5% to 12% of $u_{10N}$, depending on the wind speed. The effect of a change in measurement height by up to $5\,\mathrm{m}$ on the wind speed would be less than 1% of $u_{10N}$. Such a variation in measurement height could be caused by the uplift of the airflow that passes the ship, or due to changing buoyancy of the ship. Likewise, a deviation of the actual drag coefficient from the COARE 3.5 bulk value by 20% would lead to a change in the $u_{10N}$ estimate by about 1%.

With a 30% uncertainty in $L_*$ (assuming 20% uncertainty in $u_*$ and the temperature gradient, respectively) the uncertainty in the correction for stability ($-\frac{u_*}{\kappa} \Psi_u\left(\frac{z}{L_*}\right)$) is $\approx 1\%$ of $u_{10N}$ on average and amounts to less than 1% for 76% of the data and to less than 5% of $u_{10N}$ for 98.5% of the data, respectively.

The standard deviation of the bin averages of $\alpha_S$ ranges from 1% to 2% for most wind directions and reaches 3.5% for $-170° < D < -135°$. The values of $\alpha_S$ may, however, be biased low as they are based mainly on samples from latitudes between 60°S and 40°S, where the comparison with microwave radiometer and altimeter wind speeds indicate that ECMWF wind speeds may be on average 5% too high during the southern hemisphere summer (Stopa and Cheung, 2014). Re-evaluating the correction with 5% higher or lower ERA-5 reference wind speed leads to corrected $u_{10N}$ estimates that are different by less than 5.4% within the 16th to 84th percentile range (see Tab 1). Therefore, we estimate the common uncertainty of the flow distortion correction to be 5%.

The resulting common uncertainty estimate in the corrected $u_{10N}$ is thus given by

$$\frac{\Delta u_{10N}}{u_{10N}} \approx \sqrt{(1\%)^2 + (1\%)^2 + (5\%)^2} \approx 5\% \tag{12}$$

The flow distortion term clearly dominates the uncertainty of the corrected $u_{10N}$. Note that when the ship travels during low wind conditions the uncertainty approaches 5% of the ship velocity ($v = 8\,\mathrm{m\,s^{-1}}$) and thus $0.4\,\mathrm{m\,s^{-1}}$, which will lead to high relative uncertainties of $u_{10N}$ in these cases.

## 4.5  Local and regional variations in the ERA-5 wind speed accuracy

Figure 11 shows a map of the ship track. The marker positions indicate the ship's 6 hour average location and the marker size the magnitude of the corrected true wind speed ($u_c$), which is taken as the average of port and starboard readings. The ratio $u_M u_c^{-1}$ of the 30 meter wind speed from ERA-5 over the corrected in situ wind speed is denoted by the marker color. Along the ship track, long sections are visible where the ERA-5 wind speeds agree with the in situ wind speeds within a few percent. Large deviations of the wind speed estimates by more than 20% occur clustered along the ship track. Some of these can be linked to the vicinity of the islands that were passed by the ship. However, the ERA-5 wind speeds also deviate more than 20% in the eastern Ross sea (leg 2) and when the ship passes south west of Liberia during leg 4 (this is however not observed at the same location during leg 0).

These variations of $u_M u_c^{-1}$ show some similarities with the bias maps that Stopa and Cheung (2014), Zhang et al. (2018), and Belmonte Rivas and Stoffelen (2019) provide from comparisons of ERA-Interim and ERA-5 versus satellite-based wind speeds. Namely there is a tendency of $u_M$ to be higher than $u_c$ in the latitudes between 60°S and 40°S but lower for northern latitudes. Further the modelled wind speeds are much lower than $u_c$ in the equatorial region west of Africa.

Figure 12 presents the distribution of the observed wind speed ratios (evaluated at 1 hour resolution) for ship locations north of 40°S, between 40°S and 60°S and south of 60°S. These subsets contain 42%, 40%, and 18% of the data, respectively. For data south of 60°S, the distribution of $u_M u_c^{-1}$ is spread out the widest and features a considerable fraction of data points where $u_M$ is more than 20% lower or higher than $u_c$. Sea ice, present in this area, likely reduced the availability of satellite wind speeds for assimilation into the ECMWF model. The mean ratios and the standard error of the mean are also indicated in Fig. 12 as coloured patches (prior to the calculations the outliers were removed using the IQR-filtering). The mean ratios of the two subsets north of 40°S and south of 60°S, $0.965 \pm 0.005$ and $0.981 \pm 0.013$, respectively, do not differ significantly

from each other and are slightly lower than one. However the mean of the ratios observed for the interval 40°S to 60°S is with
1.02 ± 0.004 significantly higher than that for the other two intervals by about 5%.

## 5  Discussion

The comparison of the wind speed and direction measurements from the two anemometers of the Akademik Tryoshnikov shows
that observations are affected by airflow distortion. This comparison only allows for the detection of differences between the
wind speeds at the two anemometer locations, but not for a quantification of the absolute flow distortion bias. We use the ERA-
5 reanalysis 10 meter neutral wind speed and surface heat fluxes to calculate $u_{\mathrm{M}}$, from which we derive the modelled relative
wind speed $r_{\mathrm{M}}$. Due to the relatively low temporal and spatial resolution of the ERA-5 data, the full variability of the near
surface wind speed might be underestimated. Therefore the model predictions of the true wind speed and derived estimates of
the relative wind speed carry a relatively large uncertainty. This can, however, be reduced by the averaging over a large number
of observations. Based on the ERA-5 wind speed data we estimate a flow distortion bias in the relative wind speed ranging
from -36% to +21%. This magnitude is comparable to previous studies (e.g. Popinet et al., 2004; Landwehr et al., 2015).

For bow on wind direction the bias in both sensors is almost identical, which leads to a good agreement of the wind speed
observations from the port and starboard sensor. The agreement between measurements from two anemometers on the same
ship is often taken as an indicator for the reliability of the wind speed observation (e.g Li et al., 2013). Our observations show
that the apparent agreement of two anemometers suffering from similar flow distortion may be misleading and highlight the
case that other measures are needed to verify the quality of shipborne wind speed measurements.

When the wind speeds from the port and starboard sensor are averaged, the bias in $u_{10\mathrm{N}}$ ranges from -4% to +41% (5[th] and
95[th] percentile range) and amounts to +11% on average.

The large variability of the bias in $u_{10\mathrm{N}}$ throughout the cruise can affect correlations of independent variables with $u_{10\mathrm{N}}$,
while the mean bias can reduce the comparability to wind speed based parametrisations in the literature. As an example we
discuss the relation of $u_{10\mathrm{N}}$ to the number concentration of particles with aerodynamic diameter $D_a > 700\,\mathrm{nm}$ ($N_{700}$) which
where measured with an Aerodynamic Particle Sizer (APS) (Schmale et al., 2019b). $N_{700}$ is dominated by particles with an
aerodynamic diameter close to $1\,\mu\mathrm{m}$, which are likely to have a mean atmospheric residence time in the order of a few days
(Lewis and Schwartz, 2004). The $N_{700}$ time series has been filtered for contamination from the ship (Schmale et al., 2019a),
which effectively limits the observations to $-90° < D_{\mathrm{m}} < +90°$. Here we limit the dataset to open ocean conditions during
the legs 1, 2, and 3, where $N_{700}$ can be seen as a proxy for sea spray aerosol. Due to the long atmospheric residence time
one cannot expect a tight relation with forcing parameters of the sea spray production flux (e.g., $u_{10\mathrm{N}}$) (Lewis and Schwartz,
2004). Nonetheless, the observations from ACE may be useful to constrain sea spray emission parametrisations (manuscripts
in preparation). In Fig. 13 $N_{700}$ is plotted against $u_{10\mathrm{N}}$. The choice of $u_{10\mathrm{N}}$ calculated either from $u_{\mathrm{c}}$, $u_{\mathrm{m}}$, or using the ERA-5
data has an effect on the obtained relation and potentially deduced parametrisations. In comparison to $u_{\mathrm{c}}$, the higher values
of $u_{\mathrm{m}}$ lead to a shallower wind speed dependency. At high wind speeds $u_{10\mathrm{N}} > 12\,\mathrm{m\,s^{-1}}$, an approximately $30\%$ lower value
of $N_{700}$ would have been reported for a given wind speed. The ERA-5 data, on the other hand, does not fully resolve the

variability of $u_{10N}$ at high wind speeds and a use of $u_{10N,ERA-5}$ to parametrise $N_{700}$ would lead to the conclusion of a steeper increase with wind speed.

The along-track variation of $u_M u_c^{-1}$ has apparent similarities with the bias maps that Stopa and Cheung (2014), Zhang et al. (2018), and Belmonte Rivas and Stoffelen (2019) provide from comparisons of ERA-Interim and ERA-5 versus satellite based wind speeds. The majority of the ACE data was collected between 60°S and 40°S; thus, it is likely that a potential regional bias
for this latitude band affects the estimated correction coefficients. We have evaluated the sensitivity of the proposed correction for an assumed mean bias of 5% in the ECMWF wind speeds. For this scenario, the average bias of the corrected $u_{10N}$ estimate would amount to -3.9% on average with an IQR of 2.4%. Other uncertainties related to the wind speed profile adjustment are in the order of 1% for the majority of the data, but the uncertainty in the stability adjustment can become significant during periods of low wind speed, when the temperature gradient between the air and the sea surface is high. However, the distribution
of uncertainty in the corrected wind speed is largely reduced when compared to the variation of the bias in the $u_{10N}$ estimates that can be caused by the flow distortion.

Changes in the set-up between the four legs, which could affect the airflow pattern at the anemometer locations, could account for the variability in $\alpha_S$ and $\delta_D$ during ACE. There are no changes of the wind speed ratio of the port and starboard sensor between the legs; thus changes close to the anemometer location that would affect each sensor differently can be
excluded. The only major modification of the ship's structures between the four legs was that during leg 2 an additional crane and two containers were installed on the main deck on the starboard side of the ship. This could have potentially affected measurements in the wind sector +30° to +45°, but no evidence of this was found when $\alpha_S$ from leg 2 was compared to the other legs.

Variations of the five-minute mean roll ($-2°$ to $+2°$), mainly caused by the angle of attack and strength of the wind speed,
may explain a small fraction of the variability in the observed flow distortion. However, the variations of the five-minute mean pitch, mainly caused by changing loads, are less than $1°$ and no effect on the flow distortion pattern could be found.

## 6  Conclusions

The ACE dataset is unique in its coverage of the Southern Ocean, which is, except for a few regions, heavily under-sampled. ACE aims to establish baselines for many variables and provides the opportunity to study air-sea interaction processes in remote
regions. Many of the studied phenomena such as air-sea gas exchange or sea spray production are typically parametrised with wind speed, as they vary largely with atmospheric forcing and mixing in the surface ocean and lower atmosphere. The in situ wind speed measurement together with other meteorological variables and surface water properties thus provide an important auxiliary dataset.

On board the R/V *Akademik Tryoshnikov* in situ wind speed measurements were performed using two 2-D sonic anemome-
ters, which were integrated with an automated weather station. The relative wind speed and direction recordings from both sensors differ by up to 40% and 12°, respectively. The difference varies with the observed relative wind direction. This indicates that the measured wind speeds are affected by flow distortion caused by the ship's super structure. We also observe a

slight dependence of the wind speed ratio on the mean roll of the ship, which varied between $\pm 2°$. An influence of the pitch angle, which varied over less than $1°$ could not be found in the data.

In order to estimate the deviation of the measured wind speed and direction from the undisturbed wind field, the observed relative wind speeds are compared against a model-derived relative wind speed which was calculated from the ERA-5 wind speeds that were interpolated onto the ship's track and translated into the ship's reference system. The flow distortion bias depends on the relative wind direction and ranges between $-36\%$ to $+21\%$ and $-17°$ to $+12°$ for the relative wind speed and the relative wind direction, respectively. These observed biases are based on 1127 hours of observations (retained from

135 days at sea). Data, where the ERA-5 reanalysis may have been affected by the assimilation of observations reported by the R/V *Akademik Tryoshnikov* were excluded from this study. Data filtering and bin averaging using weighted means help to reduce the error of the bias estimates.

The biases in relative wind speed and direction can be directly used as correction factors. This correction fully preserves the high temporal resolution of the in situ wind speed measurements. In order to improve the quality of the wind speed observations

the relative wind speed and direction measurements were corrected for the estimated bias prior to the calculation of true wind and $u_{10N}$. When the wind speeds from the port and starboard sensor are averaged, the correction in $u_{10N}$ ranges from -4% to +41% (5[th] and 95[th] percentile range), and +11% on average. If the uncorrected $u_{10N}$ estimates are used with parametrisations for gas exchange or sea spray production, which are typically higher order functions of wind speed, the error propagation will lead to a much larger bias in the derived quantities.

The main advantages of the proposed weather model-based flow distortion correction for shipborne wind speeds over existing CDF based methods are the low cost of application and the option to monitor changes in the flow distortion pattern that arise from changes in the ship's superstructure over time. However, uncertainties arising from deficiencies in the small-scale and high time resolution wind characteristics of the used model reference data require careful treatment and restrict the applicability of this method to cruises that cover a wide geographical range. It should be noted that any relative bias in the weather model

wind speeds will cause an equally large or slightly lower bias in the corrected true wind speeds.

*Code and data availability.*   The underlying datasets have been or will be added to Zenodo. The code for the flow distortion analysis will be made available as git repository.

## Appendix A:  Effect of mean pitch and roll angle

The potential effect of the ship's pitch and roll on the flow distortion pattern was studied. Fig. A1 shows the effect of the

mean roll on the relative wind speed ratio $S_{m,stbd} S_{m,port}^{-1}$. The largest change in $S_{m,stbd} S_{m,port}^{-1}$ that may be attributed to the roll angle amounts to 0.07 and occurs for $D_m \approx \pm 45°$ over a change of the roll angle from $-2°$ to $+2°$. For other relative wind directions, the ratio $S_{m,stbd} S_{m,port}^{-1}$ is less sensitive to the roll angle. The five-minute mean pitch ranged from $-1.15°$ to $-1.90°$. An effect of these small variations on the flow distortion pattern could not be observed in the data. Due to the high

degree of complexity, no further attempt was made to derive a quantitative dependence of $\alpha_S$ on the roll or pitch angle. We expect the contribution to the overall uncertainty of the wind direction dependent correction factors to be small.

**Appendix B: Propagation of uncertainty in the true wind correction**

**B1    From errors in the reference wind vector to errors in the expected relative wind speed and direction**

The proposed correction relies on the calculation of the expected relative wind vector from the predicted true wind speed and the ship's velocity and heading. Figure B1 shows how the propagation of an error in $\boldsymbol{u}_\mathrm{M}$ into $S_\mathrm{M}$ and $D_\mathrm{M}$ depends on the relative wind direction and the ratio of ship velocity to true wind speed.

One should note that the relative wind direction observed on the ship will differ from the the relative angle of heading
and wind direction ($D_\mathrm{true} = \Phi_u - \Phi_H \pm 90°$) if the ship has a non zero velocity (first subplot in Figure B1). For $0 < v < u$, we find $|D| \leq |D_\mathrm{true}|$, but all directions are possible for $D$. For $v = u$, the possible relative wind directions are restricted to $-90° < D < 90°$ with $r = 0$ for $D_\mathrm{true} = \pm180°$. For $v > u$ the possible range of relative wind directions is restricted by $\sin(D) = u\,v^{-1}$, which corresponds to the situation that $\boldsymbol{u}$ and $\boldsymbol{r}$ are orthogonal and hence $\cos(180° - D_\mathrm{true}) = u\,v^{-1}$. The relative wind direction $|D|$ will decrease again for $|D_\mathrm{true}| > 180° - \arccos(u\,v^{-1})$ until $D = 0$ for $|D_\mathrm{true}| = 180°$.
On station, a relative error in $|\boldsymbol{u}_\mathrm{M}|$ results in the same relative error in $S_\mathrm{M}$ independent of the relative orientation of the ship to the wind direction and $D_\mathrm{M}$ remains unchanged.

If the ship is moving with $|\boldsymbol{v}| < |\boldsymbol{u}|$, the relative error in $|\boldsymbol{u}_\mathrm{M}|$ results in a smaller error in $S_\mathrm{M}$ for $-90° < D_\mathrm{true} < +90°$ and a magnified error for higher relative wind directions. The largest error in $D_\mathrm{M}$ occurs for $D_\mathrm{true} = \pm90°$ and the error is zero for $D_\mathrm{true} = 0°$ and $D_\mathrm{true} = 180°$.
For the special case of $|\boldsymbol{v}| = |\boldsymbol{u}_\mathrm{M}|$ we find that $|\boldsymbol{r}_\mathrm{M}| = 0$ is possible for $D_\mathrm{true} = \pm180°$ and relative uncertainty in $S_\mathrm{M}$ and $D_\mathrm{M}$ approach infinity in this case. For $D = D_\mathrm{true} = 0°$, on the other hand, the relative error in $S_\mathrm{M}$ is only half of the error in $|\boldsymbol{u}_\mathrm{M}|$.

If the ship is moving faster than the true wind speed, the relative error in $S_\mathrm{M}$ will be reduced in magnitude, but for $|D_\mathrm{true}| > 180° - \arccos(u\,v^{-1})$ a positive bias in $|\boldsymbol{u}_\mathrm{M}|$ results in a negative bias in $S_\mathrm{M}$. The resulting error in $D_\mathrm{M}$ increases with the rela-
tive wind direction, but will be smaller for the $|D_\mathrm{true}| < 180° - \arccos(u\,v^{-1})$ branch than for $|D_\mathrm{true}| > 180° - \arccos(u\,v^{-1})$ branch.

**B2    From errors in the measured relative wind speed (and direction) to errors in the true wind speed (and direction)**

If the ship is heading into the wind with $v > 0$, a flow distortion bias in $\boldsymbol{r}$ will have higher impact on $\boldsymbol{u}$, while for $\Phi_u = \Phi_H$ the relative error will be reduced, when compared to data collected while the ship is on station.

## B3 Effect of a constant relative bias in the reference wind speed

Figure B2 shows the effect of an overall reduction of the ERA-5 wind speeds by 5% on the correction factors. The sensitivity of the estimated $\alpha_S$ to such a change in the reference wind speed depends on the relative wind direction: the lowest changes (2%) occur for bow-on wind speeds and the largest changes (10%) occur for $D_m = \pm 180°$, respectively. The sensitivity of the wind direction bias estimate is very low. For the port sensor the largest changes ($\Delta \delta_D = 3°$) occur at $D_m \approx +90°$, when the sensor is in the lee of the main mast. This is the case at $D_m \approx -90°$, for the starboard sensor (not shown).

Figure B3 shows the integrated histogram of the ratio $u_\mathrm{m}\, u_\mathrm{c}^{-1}$ (compare Fig. 9 ) in comparison to the change in the corrected $u_\mathrm{c}$ for a change of the reference true wind speed by -5% ($u_{\mathrm{M}-5\%}$). On average, this would change the estimated $u_\mathrm{c}$ by -3.85%. For 27% of the data the estimate $u_\mathrm{c}$ would change by more than 5% to either larger or smaller values. A change of more than 10% would occur only for 4% of the data (ship underway at very low wind speeds).

## Appendix C: Choice of the averaging time

Figure C1 shows $S_{\mathrm{m,port}}\, S_\mathrm{M}^{-1}$ as function of $D_{\mathrm{m,port}}$ for five-minute and one-hour averages. The one-hour averages were used when at least four of the five-minute samples within the hour passed the quality control. The results are not significantly different, however, due to the lower number of the one-hour samples, reliable averages cannot be estimated for all wind direction sectors.

## Appendix D: Exclusion of intervals affected by data assimilation

Following the request of the Scientific Committee on Antarctic Research (SCAR) and the World Meteorological Organisation (WMO), the wind speed and direction observations from the R/V-*Akademik Tryoshnikov* were reported to Global Telecommuncation System (GTS) under the call sign UBXH3, while the ship location was south of $60°$S (leg 2). Uppon request ECMWF have provided a list of the time and location for which ground wind speed observations from UBXH3 where assimilated into the Integrated Forecast System (IFS) (see supplement information). The list contains 35 entries. During the remaining legs 0, 1, 3, and 4 no data from UBXH3 has been assimilated into the IFS.

ERA-5 uses 4D-Var with non-overlapping 12 hour assimilation windows, which run from 9-21 UTC and 21-9 UTC, the following day. Observations at any time within each 12 hour window can affect the analysed state over the whole 12 hours ECMWF (2016). In order to ensure that the ERA-5 estimates, which are used to estiamte the flowdistortion bias, are not affected by the assimilation of data from UBXH3, we exclude all observations within the 12 hour windows that cointain at least one instance, where the observations where assimiated.

Figure D1 shows weighted averages for $S_{\mathrm{m,port}}\, S_\mathrm{M}^{-1}$ as function of $D_{\mathrm{m,port}}$ calculated form the data with (and without) exclusion of the observations, which could have been affected by the assimilation of observations from the R/V-*Akademik Tryoshnikov*. The result for using just the subset of observations that may have been affected by the assimilation of observations

from the R/V-*Tryoshnikov* is shown aswell. Notably the estimates of $S_{\mathrm{m,port}} S_{\mathrm{M}}^{-1}$ are closer to 1 for this subset than for the remaining data.

**Appendix E: Validation of the ERA-5 SST and air temperature with insitu observations**

The atmospheric stability, which depends on the air-sea temperature gradient, affects the steepness of the wind speed profiles. In this work we account for this effect using the ERA-5 heat flux estimates. In Fig. E1 (a) ERA-5 SST estimates are scattered against insitu observations, which are combined from the calibrated temperature measuremnts of underway water intake and interpolation of Advanced Very High Resolution Radiometer onto the ship track Haumann et al. (2020). In Fig. E1 (b) the

ERA-5 estimates of the air temperature at 2 m a. s. l. are compared to the quality controled insitu observations of the air-temperature measured at 23.7 m a. s. l. provided for legs 1, 2, and 3 in Landwehr et al. (2019). The reanalysis resutls generally agree well with the observations however three events of elevated $T_a$ during legs 1 and 3 (2016-12-27, 2016-12-31 till 2017-01-01, and 2017-03-02) are not captured in the reanalysis and the cold air outbreak on 2017-01-29 is not fully resolved. This is reflected in differnt results in $\Delta T$ (see Fig. E1 (c)) and is likely related to the larger differences between $u_{\mathrm{c}}$ and $u_{\mathrm{M}}$ during

these periods (see Fig. E1 (d)).

*Author contributions.* SL did the analysis and wrote the manuscript; IT helped with handling of the ECMWF data; IT, MG, NC, and JS reviewed the manuscript and made suggestions which improved the quality of the manuscript.

*Competing interests.* The authors declare that there are no competing interests.

*Acknowledgements.* ACE-SPACE, JS, IT, and NC received funding from EPFL, the Swiss Polar Institute and Ferring Pharmaceuticals. ACE-
SPACE was carried out with additional support from the European FP7 project BACCHUS (grant agreement no. 49603445). SL received funding from the Swiss Data Science Center project c17-02. JS is the Ingvar Kamprad Chair of Extreme Environments, sponsored by Ferring Pharmaceuticals. We thank MeteoSwiss for providing access to operational ECMWF data and Michele Volpi from the Swiss Data Science Center for support in accessing the ERA-5 reanlysis data. We also thank Jenny Thomas and Carles Pina Estany for preprocessing the meteorological data from ACE.

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

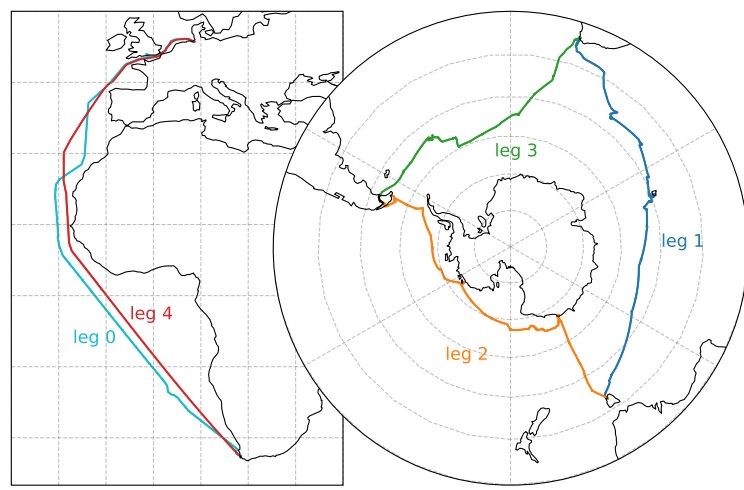

**Figure 1.** Map showing the track of the R/V-*Akademik Tryoshnikov*. The legs 0 till 4 are shown in different colors.

**Table 1.** Row one to three: percentiles of the relative bias between the uncorrected and corrected true wind speed for starboard and port anemometer separately and for the vector average of the measurements from both sensors; Row four: the percentiles of the relative bias between ERA-5 and the corrected true wind speed. Row five: same as row four but showing the relative difference between $u_c$ and $u_{c(M-5\%)}$, which was derived from ERA-5 wind speeds, which where modified to a 5% lower magnitude, in order to test the sensitivity of the correction to a bias in ERA-5. Values provided are for $u_c > 2\,\mathrm{m\,s^{-1}}$.

| | | | | | | |
|---|---|---|---|---|---|---|
| $\frac{u_m - u_c}{u_c}$ (stbd) | -0.066 | -0.001 | +0.067 | +0.302 | 0.462 | +0.121 |
| $\frac{u_m - u_c}{u_c}$ (port) | -0.059 | -0.012 | +0.077 | +0.229 | +0.353 | +0.105 |
| $\frac{u_m - u_c}{u_c}$ (avg.) | -0.041 | -0.003 | +0.063 | +0.264 | +0.407 | +0.114 |
| $\frac{u_M - u_c}{u_c}$ | -0.349 | -0.177 | 0.011 | +0.213 | +0.587 | +0.046 |
| $\frac{u_{c(M-5\%)} - u_c}{u_c}$ | -0.078 | -0.055 | -0.040 | -0.023 | -0.003 | -0.039 |

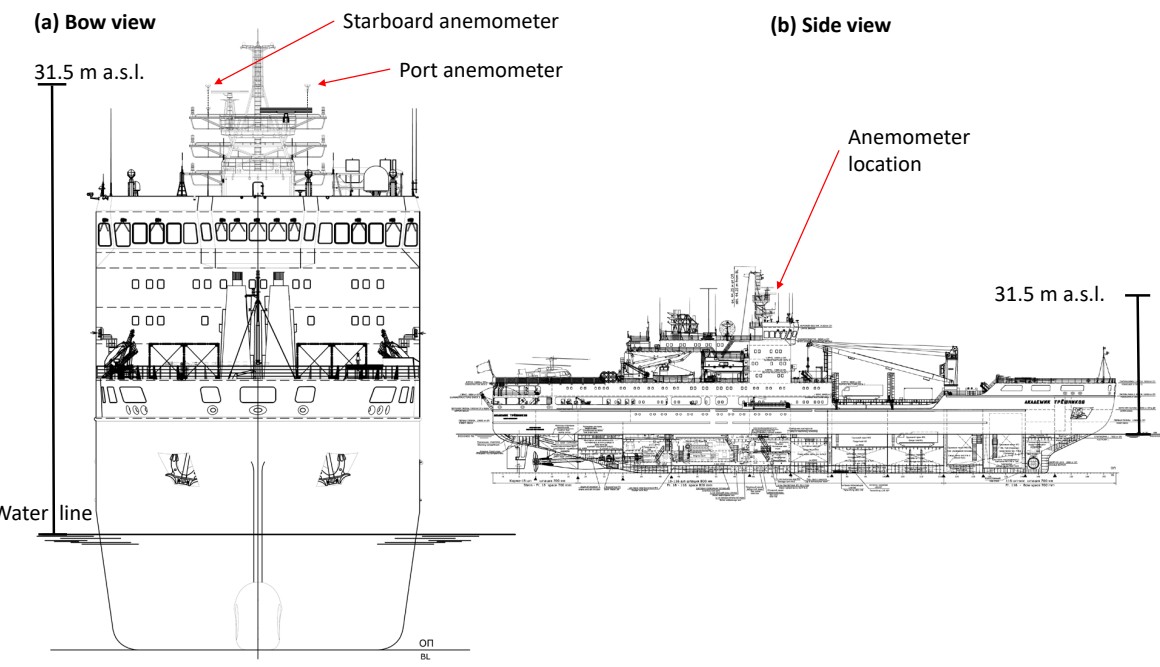

**Figure 2.** Drawing of the (a) bow and (b) side view of the R/V *Akademik Tryoshnikov*. The positions of the port and starboard anemometer are indicated by red arrows (the drawing has been modified by the author to show both anemometers mounted on the 2 meter long poles). The side view is provided in a smaller scale. Adjusted vessel plans provided by the Arctic and Antarctic Research Institute (AARI).

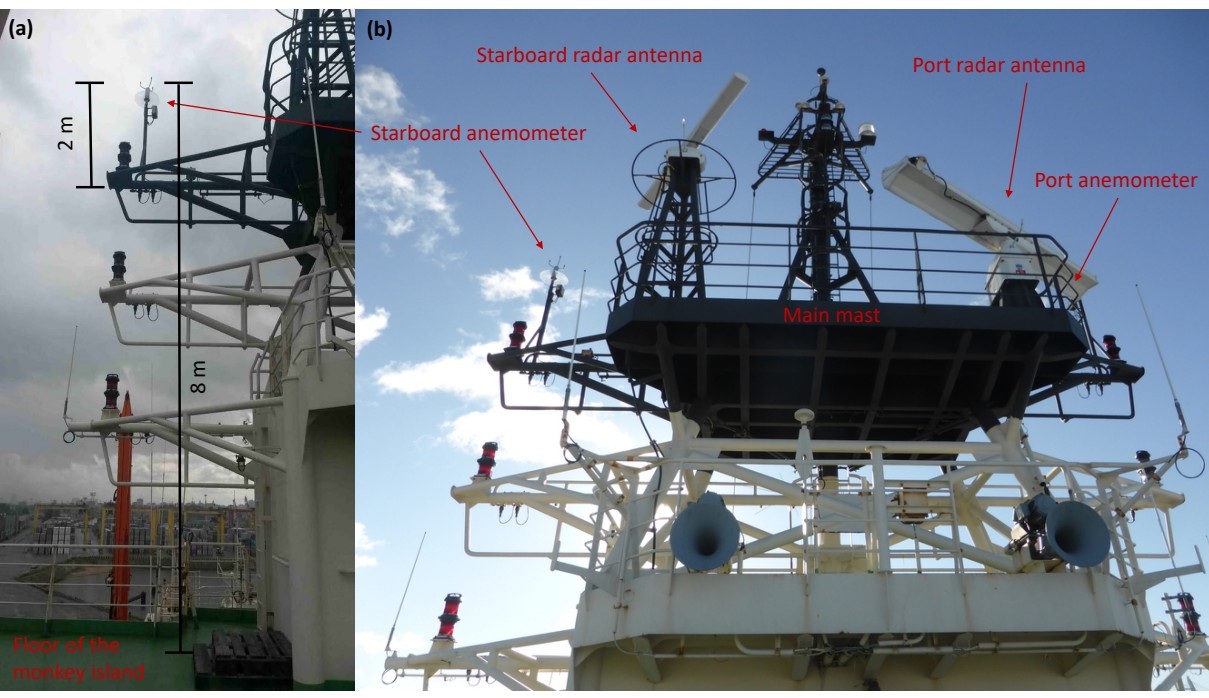

**Figure 3.** (a) and (b) annotated photographs of the set-up on the main mast. Photo credits: (a) Swiss Polar Institute (SPI); (b) Jenny Thomas.

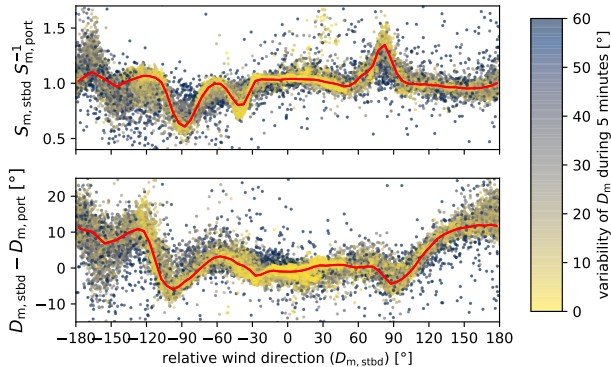

**Figure 4.** Top: ratio of the relative wind speed recorded by the starboard and port anemometers ($S_{\mathrm{m,stbd}} S_{\mathrm{m,port}}^{-1}$) as function of the relative wind direction recorded by the starboard anemometer; Bottom: difference of the relative wind directions recorded by starboard and port-side anemometer ($D_{\mathrm{m,stbd}} - D_{\mathrm{m,port}}$). The points denote 5 minute average values, the color denotes the variability of the 1 minute average wind direction within each 5 minute interval (the maximum of the variability of $D_{\mathrm{m,stbd}}$ and $D_{\mathrm{m,port}}$ ). The red line and shaded area show the average and standard error of the mean for the wind direction bins described in Sect. 3.4. The standard error of the mean was estimated via bootstrapping and amounts to $\sim 2\%$ for the relative wind speed ratio and $\sim 0.2°$ for the relative wind direction difference. (0.8% of the data reside outside the plotted range, for 4% of the data the wind direction variability is larger then $60°$)

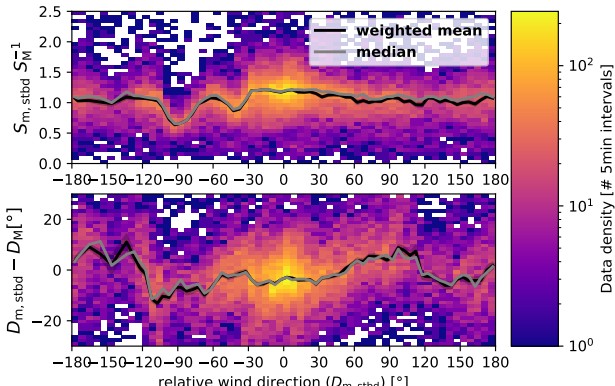

**Figure 5.** Top: Bi-dimensional histogram of $S_{\mathrm{m,stbd}}\,S_{\mathrm{M}}^{-1}$ and $D_{\mathrm{m,stbd}}$. The number of samples per ratio and wind direction bin is shown as heat map with logarithmic colour scale. The black line shows the weighted arithmetic mean calculated over each wind direction sector. The gray line shows the median, respectively. Bottom: The same as the top figure for the relative wind direction difference.

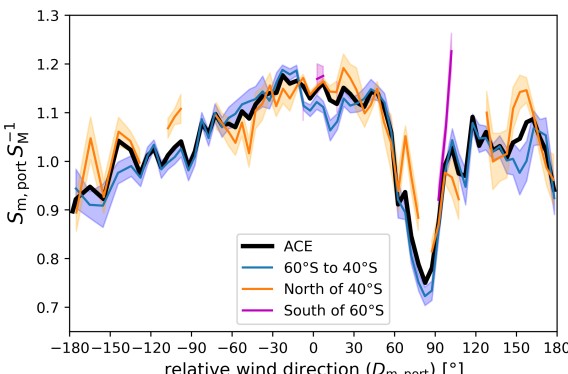

**Figure 6.** Weighted bin average ratio of $S_{\mathrm{m,port}}$ to the expected $S_{\mathrm{M}}$ (based on ERA-5) as function of $D_{\mathrm{m,port}}$. The weighted bin averages are shown for the whole ACE cruise (balck) and for the ship's location south of $60°\mathrm{S}$ (pink line), between $40°\mathrm{S}$ to $60°\mathrm{S}$ (blue line) and north of $40°\mathrm{S}$ (orange line). The shaded area denotes the standard deviation of the weighted mean.

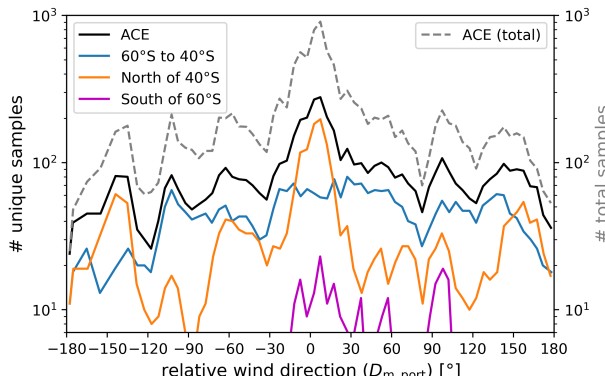

**Figure 7.** Number of "unique" observations per wind direction sector for the same subsets as shown in Fig. 6. For the number of "unique" samples in each relative wind direction sector the multiple matches of the five-minute data with the same latitude×longitude×time grid box of ERA-5 are counted only once.

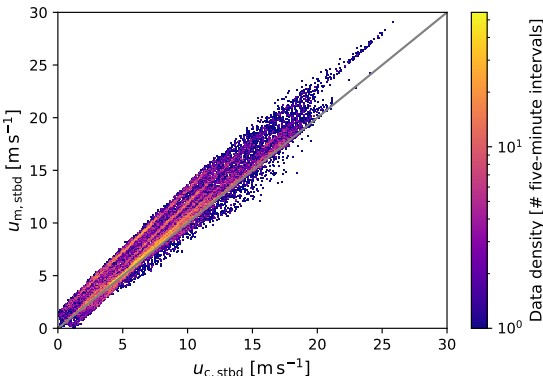

**Figure 8.** Bi-dimensional histogram of $u_{\mathrm{m,stbd}}$ against $u_{\mathrm{c,stbd}}$. Counts are provided for a $0.1\,\mathrm{m\,s^{-1}} \times 0.1\,\mathrm{m\,s^{-1}}$ wind speed resolution. The 1:1 line is indicated in gray.

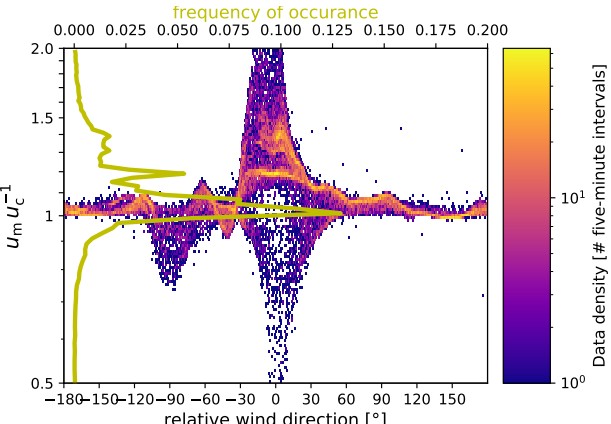

**Figure 9.** Bi-dimensional histogram of the ratio $u_{\mathrm{m,stbd}}\, u_{\mathrm{c,stbd}}^{-1}$ (the effect of the flow distortion correction on the true wind speed estimate) versus the relative wind direction $D_{\mathrm{m,stbd}}$. Counts are provided in a $1^\circ$ resolution in the wind direction and logarithmic resolution of $d\log_{10} = 0.005$ for the ratio. The yellow line shows the histogram of $u_{\mathrm{m}}\, u_{\mathrm{c}}^{-1}$ integrated over all wind directions (x-axis on top). The frequency of occurrence is provided over linear intervals of the wind speed ratio with a width of $0.02$ and plotted on the logarithmic scale of the common y-axis.

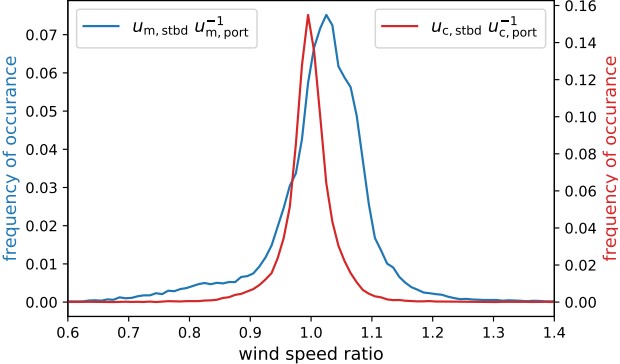

**Figure 10.** Histograms of the ratio of the true wind speed estimate from starboard and port sensor with and without the correction applied.

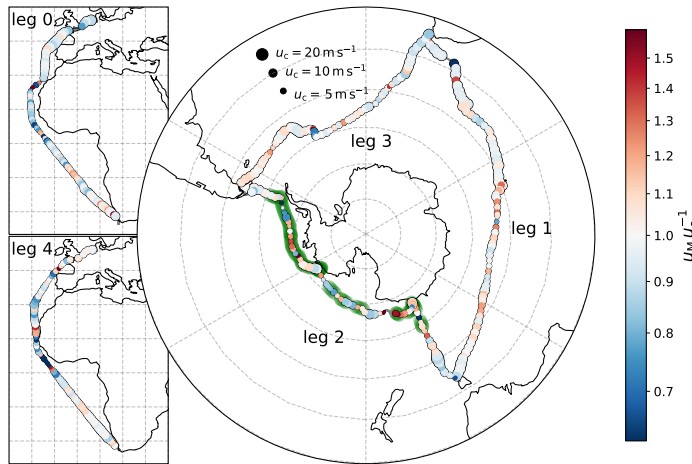

**Figure 11.** Ratio of the 31.5 meter wind speed predicted by ERA-5 over the corrected in situ wind speed along the ACE track. The marker location and maker size indicate the 6 hour mean of the ships position and of the corrected observed wind speed ($u_c$, 6 hour average of port and starboard readings), respectively. The black markers show the size for $u_c = [20, 10, 5]\,\mathrm{m\,s^{-1}}$. The the color denotes the ratio of $u_M\,u_c^{-1}$ on a logarithmic scale. The green shading denotes the part of the track, where assimilation of data reported by the Akademik Tryoshnikov may have affected the ERA-5 reanalysis results. Data with less then 50 km average distance to the nearest shore line are not shown in this plot.

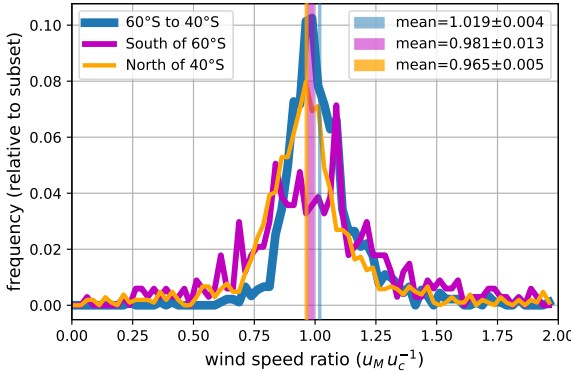

**Figure 12.** Histogram of the wind speed ratios for $u_M$ predicted by ERA-5 over $u_c$ estimated from the flow distortion corrected wind speed. The data set has been split for ship locations north of $40°\mathrm{S}$, between $40°\mathrm{S}$ and $60°\mathrm{S}$ and south of $60°\mathrm{S}$.

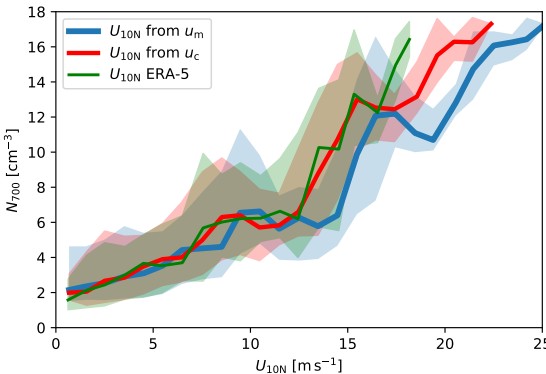

**Figure 13.** Concentration of particles with aerodynamic diameter $D_a > 700\,\mathrm{nm}$ from leg 1, 2, and 3 as function of $u_{10N}$. The lines show the median and the shaded area the IQR calculated over $1\,\mathrm{m\,s^{-1}}$ wind speed bins. Only data points where contamination form the exhaust stack emissions could be excluded, the calculated air mass back trajectories were over sea for at least 24 hours, and the distance to the nearest landmass was more than 50 km are included.

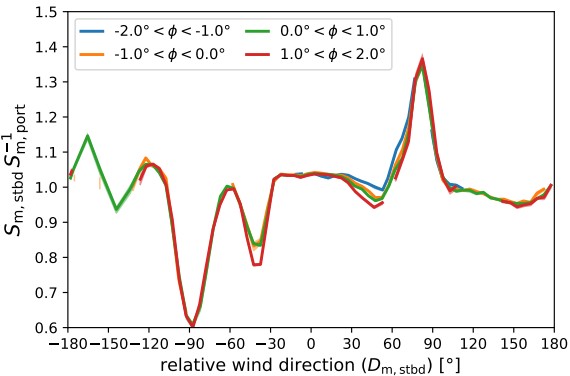

**Figure A1.** Ratio of the relative wind speed recorded by the starboard and port anemometers ($S_{\mathrm{m,stbd}} S_{\mathrm{m,port}}^{-1}$) as a function of the relative wind direction recorded by the starboard anemometer. The ratios are computed for $1°$ intervals of the ship's roll angle, which are provided in the legend.

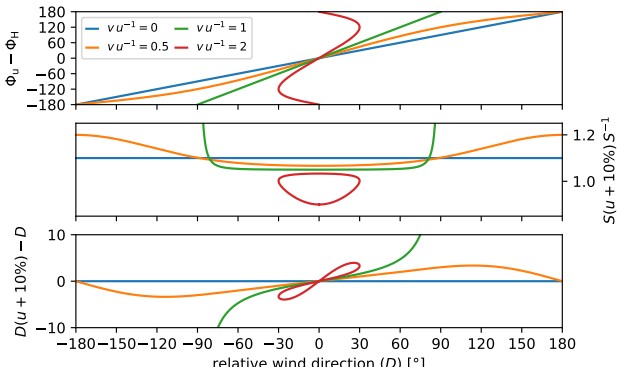

**Figure B1.** Top: Difference between true wind direction and ship heading as a function of the observable relative wind direction. Middle: Relative change in the expected relative wind speed $S_M$ for a $+10\%$ bias reference true wind speed ($u_M$) as function of the relative wind direction. Bottom: Same as middle, but showing the corresponding change in the estimated relative wind direction. The lines in different color correspond to different ratios of the ship speed and the true wind speed.

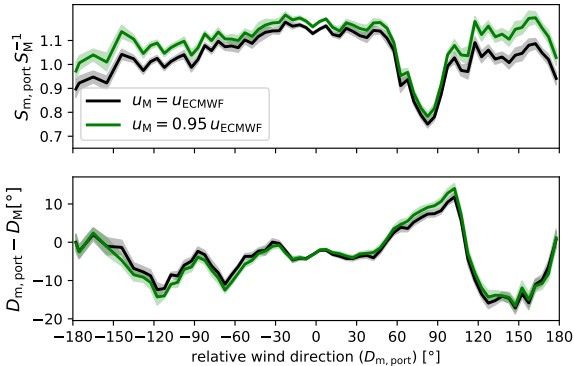

**Figure B2.** Top: weighted average ratio between measured (port anemometer) and expected relative wind speed as a function of relative wind direction. The black (green) curve shows the result for using the ERA-5 wind speeds (scaled with a factor $0.95$) as free-stream reference. The shaded areas denote the error of the mean. Bottom: the same but for the relative wind direction difference.

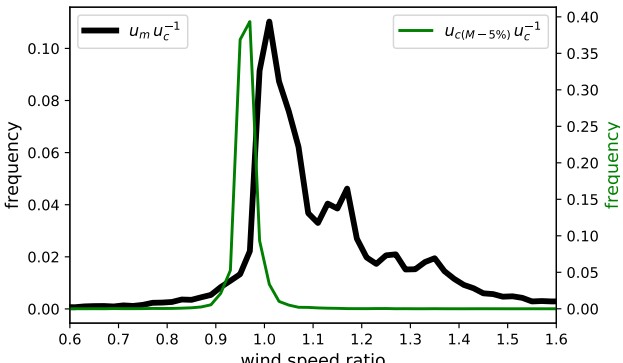

**Figure B3.** Histogram of the wind speed ratios $u_\mathrm{m}\,u_\mathrm{c}^{-1}$ (average of port and starboard measurements). Also shown is the relative change of the corrected $u_\mathrm{c}$ estimate for a change of -5% in the reference wind speed (simulating the effect of a bias in ERA-5).

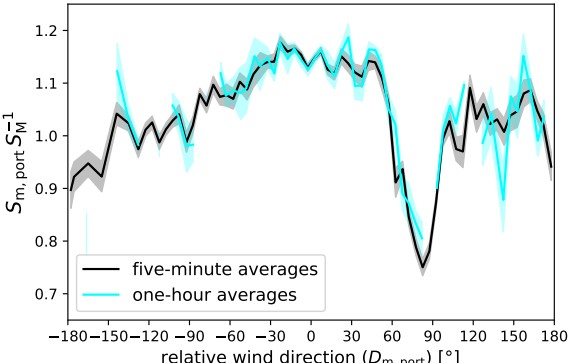

**Figure C1.** Weighted average ratio between measured (port anemometer) and expected relative wind speed as a function of the measured relative wind direction. The black (cyan) curve shows the result for five-minute and one-hour averages. The shaded areas denote the error of the mean.

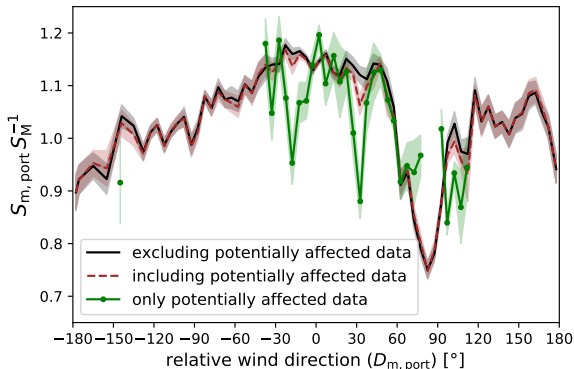

**Figure D1.** $S_{\mathrm{m,port}} S_{\mathrm{M}}^{-1}$ as function of $D_{\mathrm{m,port}}$. The curves shows the results when ERA-5 observations that could have been affected by the assimilation of the reported wind speed observations, where not included (black), included (brown), or exclusively used for the estimation of $S_{\mathrm{m,port}} S_{\mathrm{M}}^{-1}$ (green). The shaded areas denote the error of the mean.

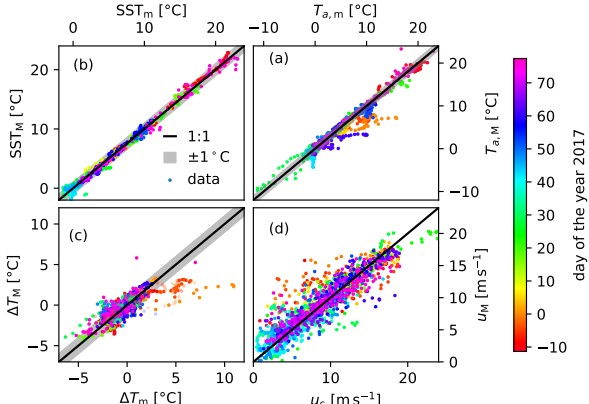

**Figure E1.** (a) ERA-5 SST against insitu observations of the surface water temperature Haumann et al. (2020); (b) ERA-5 air temperature at 2 m a. s. l. (T2M) against $T_a$ measured at 23.7 m a. s. l.; (c) ERA-5 $\Delta T$ against the observed air-water temperature difference. The transparent scatter shows data, where observations of SST where not available. Here $\Delta T_{\mathrm{m}}$ was calculated from the observed $T_a$ and the SST estimates from ERA-5; (d) $u_{\mathrm{M}}$ against $u_{\mathrm{c}}$ (31.5 meter wind speeds). This figure shows only data from legs 1, 2, and 3. All data are displayed as 1-hour average values. The color indicates the time of the observations (UTC) provided as day of the year 2017 (time difference to 2017-01-01 00:00 in days). The black lines indicate the 1:1 line and the shaded area the $\pm 1°$C range.