# Peer review of "Using global reanalysis data to quantify and correct airflow distortion bias in shipborne wind speed measurements"

_Atmospheric Measurement Techniques, 2019_

## Referee Comment (RC1) · Anonymous Referee #1 · 6 Dec 2019

General comments: Overall, this manuscript presents an intriguing evaluation of the air-flow distortion of wind measurements during the Antarctic Circumnavigation Experiment (ACE). As the authors note, identifying impacts to wind measurements from a ship's superstructure can be challenging, with CFD modeling being the most accurate, but also costly approach. Their use of reanalysis data to estimate the ship-relative winds to determine the flow bias adjustments was an interesting approach.

Overall, I found the manuscript insightful and the results were convincing. They provided a solid justification for why correcting for wind flow distortion is necessary prior to using ship wind observations to develop parameterizations of air-sea exchange processes (in their use case sea spray). The authors did a fairly good job acknowledging the limitations of their methods. I have no major concerns but point out a number of minor additions and changes that will clarify the text and figures. I believe the manuscript is suitable for publication once these minor issues are addressed.

Specific comments/suggestions:

1. Introduction, ∼line 30: The authors note that remotely sensed winds are validated using buoy wind, but should also note that ship winds have also been used to validate these systems (e.g., Bourassa, M. A., D. M. Legler, J. J. O'Brien, and S. R. Smith, 2003: SeaWinds Validation with Research Vessels. J. Geophys. Res., 108, DOI 10.1029/2001JC001028.)

2. Introduction, 5th paragraph: The authors introduce reanalyses and note the assimilation of buoys and satellite data, but neglect the fact that ship data are also assimilated to most of these models. Were any data from the ACE cruise assimilated to ERA-5? I would expect if they were, they would have been from the standard hourly bridge reports that would be contributed to the Voluntary Observing Ship scheme.

3. Introduction, 6th paragraph: Please briefly spell out the "observed effects" of pitch and roll noted in O'Sullivan et al. 2013. This will make it easier for the reader to compare your results with those from the O'Sullivan paper.

4. Introduction, 8th paragraph: Again, the authors note that reanalyses assimilate buoy and remotely sensed winds, but what about ships? On any given day there are hundreds of ships making standard weather observations over the ocean and these are archived as part of the International Comprehensive Ocean-Atmosphere Data Set (Freeman, E., S. D. Woodruff, S. J. Worley, S. J. Lubker, E. C. Kent, W. E. Angel, D. I . Berry, P. Brohan, R. Eastman, L. Gates, W. Gloeden, Z. Ji, J. Lawrimore, N. A. Rayner, G. Rosenhagen, and S. R. Smith, 2017: ICOADS Release 3.0: a major update to the historical marine climate record. Int. J. Climatol., 37, 2211–2232. doi:10.1002/joc.4775). These data are assimilated in many reanalysis products.

Please verify whether or not ship data are assimilated into ERA-5.

5. Same paragraph: Add the ACE acronym to the text where the Antarctic Circumnavigation Experiment is first mentioned.

6. Section 2, line 11 : When you mention Leg 4 in the text, it would be good to refer to the cruise tracks on a map (e.g., refer to figure 8).

7. Section 3, first paragraph: Also, a good place to refer to figure 8 cruise map.

8. Section 3.1, second paragraph. There are several items regarding the instrument set up that should be clarified a. Add more details on the different sonic anemometers used. Are the anemometer models from the same company? The acronyms/model numbers are not very useful. Do they have the same accuracy, precision, sampling rates, etc. b. How tall were the vertical poles for each anemometer? Were they the same height above the nearest deck? If you have them, photos of the installation would be great to add. c. Was the zero-reference mark on each anemometer checked before each cruise? Our experience has shown that any anemometer can come loose over time and the orientation can change. Especially if the sensor were swapped between cruise legs. Was the orientation offset between the zero-reference mark on the anemometer and the zero in the ship's coordinate reference accounted for in your calculations (see Smith et al. 1999)?

9. Section 3.2, 4th paragraph: Change "Due to the complex..." to "Due to the complexity..."

10. Same line you mention the "structures nearby the anemometers". Are these symmetric for the port and starboard anemometers? Are their differences in the upstream obstacles? Again, photos would be enlightening.

11. Section 3.4. It would be helpful to have a figure or table that quantifies the number of available observations in each relative wind sector. Also, a figure or table showing the results the sensitivity to your choice of averaging period (e.g., how much do the

results change for 5 min vs 60 min averages?).

12. Section 3.5, paragraph 3: How many unique observations were available for bias estimation after all the quality checks. The authors note 44%, but how many observations does that translate to? A table showing the number of original observations for each anemometer and the # of values removed by each test/critera would be nice.

13. Section 4.3, first paragraph: The second sentence starts "The correction tends..." Maybe I missed it, but I was not clear at this point as to how the correction was calculated. Please cross reference back to this equation earlier in the text at this point just to make this clear to the reader.

14. Section 5, 2nd paragraph: I agree completely with the authors that testing agreement between anemometers is not an indicator of their reliability. The approach presented in this paper shows promise for wider application.

15. Section 5, 3rd paragraph: the text is not clear regarding the averaging of the wind speeds. Were averages made separately for the port and starboard anemometers? Or was all data from both anemometers combined and averaged? Just a change in wording is needed to clarify. (the same wording problem exists in the conclusions, 4th paragraph).

16. Figures: All are rather small to see the details, but that may just be how they were presented to reviewers. Several plots show the relative wind direction as negative to positive – it would be helpful to label this axis with "port" and "starboard" as well to make this easy to see.

17. Figure 1: Colors are hard to differentiate (maybe a more distinct color scale would help like the one in figure 2).

18. Figure 3: Use of negative N in the latitude labels is confusing. In the text you call this south (S) latitude. Please do the same in the figures to be consistent with the text.

19. Figure 8: Sort the data by the magnitude of the difference and plot with the largest

ratios (both + and -) on top. At present, some of the smaller ratios are plotted over the larger ratios, thus the plot underestimates the differences.

20. Figure 10: Please clarify the meaning of the lines versus the shading in this figure. I assume the line is the median, and the shading the IQR, but state this in the caption.

———————————————

---

## Referee Comment (RC2) · Anonymous Referee #2 · 11 Dec 2019

General

This manuscript presents the correction of shipborne wind observations, which are biased by flow distortion. With the use of reanalysis data these biases can be quantified and the observations can be corrected subsequently. Eventually, the uncertainty of the observations after correction is in the range of the uncertainty of the reanalysis product.

This work is an important contribution, because in-situ wind speed and direction observations on the open ocean are still rare. Therefore, the existing observations from research vessels, buoys, and other platforms need a critical review since they are used for a variety of purposes like scatterometer calibration, model validation and estimates

of air-sea exchange, which are often parameterized with wind speed.

The authors clearly explain their motivation and their method. They use ERA-5 data to fill the gap between observations and to detect and quantify flow distortion, which itself is dependent on the relative wind direction. The ranges of the biases are large for both, relative wind speed and direction. Converting this into true winds, a large error is estimated, which is reduced after applying the bias correction estimated from the flow distortion. As a major result the authors show a final dependence of the corrected bias to the used ERA-5 product. Problems and limits of the approach are discussed and illustrated.

I suggest a minor revision as the manuscript is clearly structured and the scientific workflow is properly described. However, I have some minor specific comments, which are described in detail below.

Specific comments

There are some inconsistencies in the labels and/or captions of the figures (cf. technical comments to the figures). Please elaborate generally: Whenever data from one sensor are shown, make sure it is stated consistently in the text/labels/caption.

Technical comments/suggestions

Page 2/line 26-29: Just a comment. It's true, that buoys are the backbone for validation of other wind products. The impact of flow distortion is smaller compared to ships, right. However, flow distortion is an issue for buoys, too. Similar to ships this flow distortion is highly dependent from the structure on the buoy. The problem is that usually it is either not recognized or one is not able to estimate this effect due to the lack of redundant observations. Emond et al., 2012 and Bigorre et al., 2013 extensively studied these effects, which can be on the order of 5-10% of the observed wind speeds.

3/21-22: See above comment on buoy flow distortion biases.

3/26: Twice per day is rather good from a global point of view. There is even the

RapidSCAT program, which deals with the diurnal cycle in wind speed. However, for your purposes it's still small.

4/9: Please introduce the abbreviation ACE first (perhaps on page 3/line 31 or whenever it shows up first after the abstract).

4/16: The mentioned study describes altimeter and radiometer observations. They don't deal with scatterometers, do they?

5/3 (first paragraph): Even though it shows up later in figure 8, a map at this part of the paper might help the reader to follow.

5/24: How often does this happen? Can you give an example? Parking of the ship?

6/14: No SSTs from the weather station? What do you mean with 'not yet available'?

6/23: Replace "form" with "from".

9/3: I'd like to read here a number(or a ratio) of how many data are finally used for the estimation of flow distortion parameters. Just to get an impression. I calculated 40.5% (of all 'raw' data). Is that right?

9/9: It is five-minute average? Or five (times) minute-averages? I'd suggest to use five-minute, i.e. with a hyphen, and continue this throughout the paper.

9/22: A function of which relative wind direction? As you've shown before the measurements between the two sensors can differ strikingly. Please clarify. (See also comment to figure 3)

10/31: It looks overcorrected in figure 7, meaning that your peak is now below the ratio 1.0? Any comment on that?

11/4: Unclear formulation "could be caused the uplift". You mean "caused by …". Please clarify.

11/19: What other sources of uncertainty can play a role for u10N?

Fig. 1: Caption: Remove one 'the' in the second sentence.

Fig.3: I'm a bit confused. In the text you describe model against starboard, which is also true for the labels. In the caption you describe model/port ratio and difference. Which one is true? What is on the y-axis, port or starboard? Please clarify.

Additional references

Bigorre, S. P., Weller, R. A., Edson, J. B., & Ware, J. D. (2013). A surface mooring for air–sea interaction research in the Gulf Stream. Part II: Analysis of the observations and their accuracies. Journal of Atmospheric and Oceanic Technology, 30(3), 450-469.

Emond, M., Vandemark, D., Forsythe, J., Plueddemann, A. J., & Farrar, J. T. (2012). Flow distortion investigation of wind velocity perturbations for two ocean meteorological platforms. Woods Hole Oceanographic Institution.

---

## Author Response (AR1)

**Response to Referee #2**

Many thanks to Referee #2 for her/his constructive and supportive comments, which we have kept in *italic* and labelled as referee comment (RC). We provide our author replies (AR) below.

**General**: *This manuscript presents the correction of shipborne wind observations, which are biased by flow distortion. With the use of reanalysis data these biases can be quantified and the observations can be corrected subsequently. Eventually, the uncertainty of the observations after correction is in the range of the uncertainty of the reanalysis product.*
*This work is an important contribution, because in-situ wind speed and direction observations on the open ocean are still rare. Therefore, the existing observations from research vessels, buoys, and other platforms need a critical review since they are used for a variety of purposes like scatterometer calibration, model validation and estimates of air-sea exchange, which are often parameterized with wind speed.*
*The authors clearly explain their motivation and their method. They use ERA-5 data to fill the gap between observations and to detect and quantify flow distortion, which itself is dependent on the relative wind direction. The ranges of the biases are large for both, relative wind speed and direction. Converting this into true winds, a large error is estimated, which is reduced after applying the bias correction estimated from the flow distortion. As a major result the authors show a final dependence of the corrected bias to the used ERA-5 product. Problems and limits of the approach are discussed and illustrated.*
*I suggest a minor revision as the manuscript is clearly structured and the scientific workflow is properly described. However, I have some minor specific comments, which are described in detail below.*

**Authors response**: We thank Referee #1 for her/his careful review and the helpful comments.

**Specific comments**: *There are some inconsistencies in the labels and/or captions of the figures (cf. technical comments to the figures). Please elaborate generally: Whenever data from one sensor are shown, make sure it is stated consistently in the text/labels/caption.*

**Authors response**: Thank your for this note. We have reviewed the manuscript to improve the consistency.

**Technical comments/suggestions:**

**RC 1:** *Page 2/line 26-29: Just a comment. It's true, that buoys are the backbone for validation of other wind products. The impact of flow distortion is smaller compared to ships, right. However, flow distortion is an issue for buoys, too. Similar to ships this flow distortion is highly dependent from the structure on the buoy. The problem is that usually it is either not recognized or one is not able to estimate this effect due to the lack of redundant observations. Emond et al., 2012 and Bigorre et al., 2013 extensively studied these effects, which can be on the order of 5-10% of the observed wind speeds.*

**AR 1:** Thank you for this comment. We have modified our statement to specify the range of flow distortion errors in buoy observations found by Emond et al., 2012 and Bigorre et al., 2013: "For buoys, the ratio of the sensor's height above the main structure to the dimension of the structure is much higher, so that airflow distortion is typically lower, in the order of 5% to 10% (e.g. Emond et al., 2012; Bigorre et al., 2013)."

**RC 2:** *3/21-22: See above comment on buoy flow distortion biases.*

**AR 2:** The bias in the buoy wind speeds may explain some of the scatter in (Landwehr et al., 2015, Fig. 5), unfortunately I was not aware of the study of Emond et al., (2012) at that time. However 5% are small compared to the errors found in the wind speeds measured at the temporary bow mast of the *Saramiento de Gamboa*.

**RC 3:** *3/26: Twice per day is rather good from a global point of view. There is even the RapidSCAT program, which deals with the diurnal cycle in wind speed. However, for your purposes it's still small.*

**AR 3:** Yes indeed.

**RC 4:** *4/9: Please introduce the abbreviation ACE first (perhaps on page 3/line 31 or whenever it shows up first after the abstract).*

**AR 4:** Thanks. We have added the introduction of the abbreviation as suggested.

**RC 5:** *4/16: The mentioned study describes altimeter and radiometer observations. They don't deal with scatterometers, do they?*

**AR 5:** Yes, Yound and Donelan, (2018) describe biases in observation from altimeter and radiometer sensors but not from scatterometers. We have corrected the mistake.

**RC 6:** *5/3 (first paragraph): Even though it shows up later in figure 8, a map at this part of the paper might help the reader to follow.*

**AR 6:** We have added a new figure showing the ship track during the four legs.

**RC 7:** *5/24: How often does this happen? Can you give an example? Parking of the ship?*

**AR 7:** The difference between the five-minute average course and the five-minute average heading (vector averages) is larger than $10°$ for about 22% of the samples. This occurs when the ship is on station, so that the GPS velocity is too small to estimate a reliable course.

**RC 8:** *6/14: No SSTs from the weather station? What do you mean with 'not yet available'?*

**AR 8:** The calibration and quality control of the SST measurements from the underway system was ongoing at the time of submission. It has been finalized in the meantime and we are now using the in situ observations combined with SST estimates from remote sensing, instead of the ERA-5 output.

**RC 9:** *6/23: Replace "form" with "from".*

**AR 9:** Thanks!

**RC 10:** *9/3: I'd like to read here a number(or a ratio) of how many data are finally used for the estimation of flow distortion parameters. Just to get an impression. I calculated 40.5(of all 'raw' data). Is that right?*

**AR 10:** For the port and starboard sensor 13353 and 13529 five-minute samples pass all quality checks the are 35% of the originally available data. Note that about 10 days of data from leg 0 where additional available and that the removal of intervals where ERA-5 might have been affected by the assimilation of wind speed and direction data from the Akademik Tryoshinkov lead to a removal of about 277 hours of observations (see responses to comments 2 and 12 from Referee #1). We have added this information in the text.

**RC 11:** *9/9: It is five-minute average? Or five (times) minute-averages? I'd suggest to use five-minute, i.e. with a hyphen, and continue this throughout the paper.*

**AR 11:** Thanks for the suggestion, we have changed all occurrences.

**RC 12:** *9/22: A function of which relative wind direction? As you've shown before the measurements between the two sensors can differ strikingly. Please clarify. (See also comment to figure 3)*

**AR 12:** The relative wind direction measured by the starboard sensor. We have now clarified this in the text.

**RC 13:** *10/31: It looks overcorrected in figure 7, meaning that your peak is now below the ratio 1.0? Any comment on that?*

**AR 13:** The peak of the histogram of the corrected wind speed ratios is at 0.995. Considering the with of the ratio bins used (0.01), this is very close to 1.

**RC 14:** *11/4: Unclear formulation "could be caused the uplift". You mean "caused by ...". Please clarify.*

**AR 14:** Yes, the "by" was missing.

**RC 15:** *11/19: What other sources of uncertainty can play a role for $u_{10N}$?*

**AR 15:** Here we considered the accuracy of the measurement height, the influence of the atmospheric stability on the shape of the wind speed profile, and uncertainties in the drag coefficient. Each of these uncertainty sources contributes less than 1% to the relative uncertainty of $u_{10N}$.

**RC 16:** *Fig. 1: Caption: Remove one 'the' in the second sentence.*

**AR 16:** Thanks!

**RC 17:** *Fig.3: I'm a bit confused. In the text you describe model against starboard, which is also true for the labels. In the caption you describe model/port ratio and difference. Which one is true? What is on the y-axis, port or starboard? Please clarify.*

**AR 17:** Fig. 3 shows data from the starboard sensor. The caption was wrong and has been corrected.

References

Bigorre, S. P., Weller, R. A., Edson, J. B., & Ware, J. D. (2013). A surface mooring for air–sea interaction research in the Gulf Stream. Part II: Analysis of the observations and their accuracies. Journal of Atmospheric and Oceanic Technology, 30(3), 450-469.

Emond, M., Vandemark, D., Forsythe, J., Plueddemann, A. J., & Farrar, J. T. (2012). Flow distortion investigation of wind velocity perturbations for two ocean meteorological platforms. Woods Hole Oceanographic Institution.

Young, I. R. and Donelan, M. A. (2018): On the Determination of Global Ocean Wind and Wave Climate from Satellite Observations, Remote Sensing of Environment, 215, 228–241, https://doi.org/10.1016/j.rse.2018.06.006, 2018.

**Response to Referee #1**

Many thanks to Referee #1 for her/his constructive and supportive comments, which we have kept in *italic* and labelled as referee comment (RC). We have adjusted the figure numbers in the original comments of Referee #1 to match the submitted manuscript. We provide our author replies (AR) below:

**General comments**: *Overall, this manuscript presents an intriguing evaluation of the air-flow distortion of wind measurements during the Antarctic Circumnavigation Experiment (ACE). As the authors note, identifying impacts to wind measurements from a ship's superstructure can be challenging, with CFD modeling being the most accurate, but also costly approach. Their use of reanalysis data to estimate the ship-relative winds to determine the flow bias adjustments was an interesting approach. Overall, I found the manuscript insightful and the results were convincing. They provided a solid justification for why correcting for wind flow distortion is necessary prior to using ship wind observations to develop parameterizations of air-sea exchange processes (in their use case sea spray). The authors did a fairly good job acknowledging the limitations of their methods. I have no major concerns but point out a number of minor additions and changes that will clarify the text and figures. I believe the manuscript is suitable for publication once these minor issues are addressed.*

**Authors response**: We thank Referee #1 for pointing out the value of this contribution and her/his careful review.

**Specific comments/suggestions:**

**RC 1:** *Introduction, line 30: The authors note that remotely sensed winds are validated using buoy wind, but should also note that ship winds have also been used to validate these systems (e.g., Bourassa, M. A., D. M. Legler, J. J. O'Brien, and S. R. Smith, 2003: SeaWinds Validation with Research Vessels. J. Geophys. Res., 108, DOI 10.1029/2001JC001028.)*

**AR 1:** We have extended the sentence to "...and from voluntarily observing ships (e.g., Bourassa et al. 2003)"

**RC 2:** *Introduction, 5th paragraph: The authors introduce reanalyses and note the assimilation of buoys and satellite data, but neglect the fact that ship data are also assimilated to most of these models. Were any data from the ACE cruise assimilated to ERA-5? I would expect if they were, they would have been from the standard hourly bridge reports that would be contributed to the Voluntary Observing Ship scheme.*

**AR 2:** Indeed wind speed observations from ships are still used in the data assimilation for ECMWF weather forecast model. In order to acknowledge this fact, we have added ships in the list of the in situ observations in page 3 line 3. The *Akademik Tryoshnikov* reported daily telegrams to the Arctic and Antarctic Research Institute (AARI). Following the request of the Scientific Committee on Antarctic Research (SCAR) and the World Meteorological Organisation (WMO),

the observations were also reported to Global Telecommuncation System (GTS) under the call sign UBXH3. ECMWF have provided us with a list of the time and location for which ground wind speed observations from UBXH3 where assimilated into the Integrated Forecast System (IFS) (see supplement information). The list contains 35 entries. The data from intervals close to these time stamps cluster around ratio of $S_\mathrm{m} S_\mathrm{M}^{-1} \approx 1$, however, with considerable scatter. To avoid any feedback in the bias correction, we excluded all observations that where within a 4D-VAR assimilation window (09:00 UTC to 21:00 UTC or 21:00 UTC to 09:00 UTC on the following day) during which data from UBXH3 was assimilated into the IFS. This leads to a removal of 277 hours of data.

**RC 3:** *Introduction, 6th paragraph: Please briefly spell out the "observed effects" of pitch and roll noted in O'Sullivan et al. 2013. This will make it easier for the reader to compare your results with those from the O'Sullivan paper.*

**AR 3:** We changed the sentence to: "In the results of their CFD simulation, O'Sullivan et al. (2013) observed changes in the relative wind speed bias in dependence of the pitch and roll of the ship as well as the magnitude of the relative wind speed."

**RC 4:** *Introduction, 8th paragraph: Again, the authors note that reanalyses assimilate buoy and remotely sensed winds, but what about ships? On any given day there are hundreds of ships making standard weather observations over the ocean and these are archived as part of the International Comprehensive Ocean-Atmosphere Data Set (Freeman, E., S. D. Woodruff, S. J. Worley, S. J. Lubker, E. C. Kent, W. E. Angel, D. I . Berry, P. Brohan, R. Eastman, L. Gates, W. Gloeden, Z. Ji, J. Lawrimore, N. A. Rayner, G. Rosenhagen, and S. R. Smith, 2017: ICOADS Release 3.0: a major update to the historical marine climate record. Int. J. Climatol., 37, 2211– 2232. doi:10.1002/joc.4775). These data are assimilated in many reanalysis products. Please verify whether or not ship data are assimilated into ERA-5.*

**AR 4:** Also here we added ships to the list of data in situ observing platforms.

**RC 5:** *Same paragraph: Add the ACE acronym to the text where the Antarctic Circumnavigation Experiment is first mentioned.*

**AR 5:** Thanks! The acronym has been added.

**RC 6:** *Section 2, line 11 : When you mention Leg 4 in the text, it would be good to refer to the cruise tracks on a map (e.g., refer to figure 8).*

**AR 6:** We added a reference to Fig. 4 (a) in Young and Donelan (2018)

**RC 7:** *Section 3, first paragraph: Also, a good place to refer to figure 8 cruise map.*

**AR 7:** Following the request of a map in the methods section, we added an additional Figure to

 show the track of ACE during legs 0–4 and keep Figure 8 in the results section.

**RC 8:** *Section 3.1, second paragraph. There are several items regarding the instrument set up that should be clarified a. Add more details on the different sonic anemometers used. Are the anemometer models from the same company? The acronyms/model numbers are not very useful. Do they have the same accuracy, precision, sampling rates, etc. b. How tall were the vertical poles for each anemometer? Were they the same height above the nearest deck? If you have them, photos of the installation would be great to add. c. Was the zero-reference mark on each anemometer checked before each cruise? Our experience has shown that any anemometer can come loose over time and the orientation can change. Especially if the sensor were swapped between cruise legs. Was the orientation offset between the zero-reference mark on the anemometer and the zero in the ship's coordinate reference accounted for in your calculations (see Smith et al. 1999)?*

**AR 8:** a) We added the information that both anemometer are distributed by Vaisala. The model numbers allow to quickly find manuals and further references. The sampling rate was 1/3 seconds, the accuracies (1% and 2% for WMT702 and WS425 and 2° for both) are mentioned in Section 3.3. b) The poles were about 2 m tall and where at the same height (8 meter) above the nearest deck. Please note that the height of the anemometer was indeed 31.5 and not 30.5 m a.s.l. We have also corrected the annotation of the drawings in Figure 1. However, this small change does not affect our results. We have now added all this information in Section 3.1. and provide an additional figure with photographs of the set-up.
c) To our knowledge all sensors where checked to prior the cruise. We have no information that there would have been an offset between the zero-reference and the ship's main axis. Sensors where not swapped or relocated in between the cruise legs. Considering symmetry of the wind speed ratios and direction differences around the ship's main axis, we can exclude any major offsets.

**RC 9:** *Section 3.2, 4th paragraph: Change "Due to the complex. . . " to "Due to the complexity. . . "*

**AR 9:** We have corrected this mistake.

**RC 10:** *Same line you mention the "structures nearby the anemometer". Are these symmetric for the port and starboard anemometers? Are their differences in the upstream obstacles? Again, photos would be enlightening.*

**AR 10:** The two radar antenna mounted on the top platform of the main mast (see photographs) are the two main features that introduce asymmetry to the set-up. We do now elaborate on this in Section 3.1.

**RC 11:** *Section 3.4. It would be helpful to have a figure or table that quantifies the number of available observations in each relative wind sector. Also, a figure or table showing the results the sensitivity to your choice of averaging period (e.g., how much do the results change for 5 min vs 60 min averages?).*

**AR 11:** We currently provide the number of "unique" samples per wind direction bin in the bottom

panel in Figure 4. To make the graphics more readable we split Figure 4 into two figures. Besides the number of "unique" samples we do now also provide the total number of samples per wind sector in the figure. We have added an appendix figure (Figure C1) to show $S_{\mathrm{m,port}} S_{\mathrm{M}}^{-1}$ as well as $D_{\mathrm{m,port}} - D_{\mathrm{M}}$ for 5 min and 60 min averages. The results do not change significantly for 5-minute vs 60-minute averages however due to the lower number of samples for 60-minute averages, we cannot find a reliable average ratio for some of the wind sectors.

**RC 12:** *Section 3.5, paragraph 3: How many unique observations were available for bias estimation after all the quality checks. The authors note 44%, but how many observations does that translate to? A table showing the number of original observations for each anemometer and the # of values removed by each test/criteria would be nice.*

**AR 12:** We avail of 37835 and 37925 five-minute average observations from the port and starboard sensor. For the port and starboard sensor 15209 and 15397 five-minute average observations remain after the quality controls (including the removal of data, which are potentially affected by the assimilation of data insitu observations into the IFS) these are 35% of the available data. After the IQR-based outlier removal, 13353 and 13529 observations remain, which amounts to 35% of the originally available data that pass all quality control measures and are used to derive the flow distortion correction factors. For the port sensor, the number of unique samples is provided in Figure 4 there is not much differnce for the starboad sensor. We have now added the number of available and finally used samples in the text.

**RC 13:** *Section 4.3, first paragraph: The second sentence starts "The correction tends..." Maybe I missed it, but I was not clear at this point as to how the correction was calculated. Please cross reference back to this equation earlier in the text at this point just to make this clear to the reader.*

**AR 13:** We changed the sentence to: "The correction of the measured wind vector via Eq. (9) tends to reduce the true wind speed but the magnitude of the correction varies by more than $5\,\mathrm{m\,s^{-1}}$."

**RC 14:** *Section 5, 2nd paragraph: I agree completely with the authors that testing agreement between anemometers is not an indicator of their reliability. The approach presented in this paper shows promise for wider application.*

**AR 14:** Thank you for the acknowledgement. We will make the code available to facilitate the application of this method.

**RC 15:** *Section 5, 3rd paragraph: the text is not clear regarding the averaging of the wind speeds. Were averages made separately for the port and starboard anemometers? Or was all data from both anemometers combined and averaged? Just a change in wording is needed to clarify. (the same wording problem exists in the conclusions, 4th paragraph).*

**AR 15:** We change this to "When the wind speed measurements from the port and starboard sensor are averaged, ..."

**RC 16:** *Figures: All are rather small to see the details, but that may just be how they were presented to reviewers. Several plots show the relative wind direction as negative to positive – it would be helpful to label this axis with "port" and "starboard" as well to make this easy to see.*

**AR 16:** We add the following sentence to Section 3.1. "Where $D$ is used as x-axis in the figures, we have values reaching from $-180°$ to $+180°$ in order to create a panorama. Negative values of $D$ denote wind from the port side and positive values from the starboard side, respectively." We find this display very intuitive and don't see the necessity to add further annotation to the figures.

**RC 17:** *Figure 2: Colors are hard to differentiate (maybe a more distinct color scale would help like the one in figure 3).*

**AR 17:** We changed Figure 2 to a different color scale and increased the marker size. The main point of this figure is to show the variability in $S_m S_M^{-1}$ and $D_m - D_M$, and to highlight that this variability is mostly related to direction changes of the ship within the averaging interval.

**RC 18:** *Figure 4: Use of negative N in the latitude labels is confusing. In the text you call this south (S) latitude. Please do the same in the figures to be consistent with the text.*

**AR 18:** We changed the labels in Figures 4 and 9 to "60°S to 40°S", "North of 40°S", and "South of 60°S".

**RC 19:** *Figure 8: Sort the data by the magnitude of the difference and plot with the largest ratios (both + and -) on top. At present, some of the smaller ratios are plotted over the larger ratios, thus the plot underestimates the differences.*

**AR 19:** The main point of this figure is to highlight regions and prolonged time periods, when the wind speeds differ systematically. The data are plotted consecutively in time so no bias towards larger or smaller differences is introduced. We choose to indicate the magnitude of the true wind speed with the marker size to highlight discrepancies for high wind speeds that are more relevant to air-sea gas-exchange and sea spray emission. The issue of entirely overlapping data points occurs only for periods where the ships was moored, this could only be fixed by presenting the data as time series. We have reduced the overall marker size in order to reduce the overlap between neighbouring observations.

**RC 20:** *Figure 10: Please clarify the meaning of the lines versus the shading in this figure. I assume the line is the median, and the shading the IQR, but state this in the caption.*

**AR 20:** Indeed the lines are the median values while the shading indicates the IQR. We have modified the caption to clarify this.

Atmos. Meas. Tech. Discuss.,
doi:10.5194/amt-2019-366-RC2, 2019

[Figure]

This manuscript presents the correction of shipborne wind observations, which are biased by flow distortion. With the use of reanalysis data these biases can be quantified and the observations can be corrected subsequently. Eventually, the uncertainty of the observations after correction is in the range of the uncertainty of the reanalysis product.

This work is an important contribution, because in-situ wind speed and direction observations on the open ocean are still rare. Therefore, the existing observations from research vessels, buoys, and other platforms need a critical review since they are used for a variety of purposes like scatterometer calibration, model validation and estimates

of air-sea exchange, which are often parameterized with wind speed.

The authors clearly explain their motivation and their method. They use ERA-5 data to fill the gap between observations and to detect and quantify flow distortion, which itself is dependent on the relative wind direction. The ranges of the biases are large for both, relative wind speed and direction. Converting this into true winds, a large error is estimated, which is reduced after applying the bias correction estimated from the flow distortion. As a major result the authors show a final dependence of the corrected bias to the used ERA-5 product. Problems and limits of the approach are discussed and illustrated.

I suggest a minor revision as the manuscript is clearly structured and the scientific workflow is properly described. However, I have some minor specific comments, which are described in detail below.

Specific comments

There are some inconsistencies in the labels and/or captions of the figures (cf. technical comments to the figures). Please elaborate generally: Whenever data from one sensor are shown, make sure it is stated consistently in the text/labels/caption.

Technical comments/suggestions

Page 2/line 26-29: Just a comment. It's true, that buoys are the backbone for validation of other wind products. The impact of flow distortion is smaller compared to ships, right. However, flow distortion is an issue for buoys, too. Similar to ships this flow distortion is highly dependent from the structure on the buoy. The problem is that usually it is either not recognized or one is not able to estimate this effect due to the lack of redundant observations. Emond et al., 2012 and Bigorre et al., 2013 extensively studied these effects, which can be on the order of 5-10% of the observed wind speeds.

3/21-22: See above comment on buoy flow distortion biases.

3/26: Twice per day is rather good from a global point of view. There is even the

[Figure]

RapidSCAT program, which deals with the diurnal cycle in wind speed. However, for your purposes it's still small.

4/9: Please introduce the abbreviation ACE first (perhaps on page 3/line 31 or whenever it shows up first after the abstract).

4/16: The mentioned study describes altimeter and radiometer observations. They don't deal with scatterometers, do they?

5/3 (first paragraph): Even though it shows up later in figure 8, a map at this part of the paper might help the reader to follow.

5/24: How often does this happen? Can you give an example? Parking of the ship?

6/14: No SSTs from the weather station? What do you mean with 'not yet available'?

6/23: Replace "form" with "from".

9/3: I'd like to read here a number(or a ratio) of how many data are finally used for the estimation of flow distortion parameters. Just to get an impression. I calculated 40.5% (of all 'raw' data). Is that right?

9/9: It is five-minute average? Or five (times) minute-averages? I'd suggest to use five-minute, i.e. with a hyphen, and continue this throughout the paper.

9/22: A function of which relative wind direction? As you've shown before the measurements between the two sensors can differ strikingly. Please clarify. (See also comment to figure 3)

10/31: It looks overcorrected in figure 7, meaning that your peak is now below the ratio 1.0? Any comment on that?

11/4: Unclear formulation "could be caused the uplift". You mean "caused by ...". Please clarify.

11/19: What other sources of uncertainty can play a role for u10N?

[Figure]

Fig. 1: Caption: Remove one 'the' in the second sentence.

Fig.3: I'm a bit confused. In the text you describe model against starboard, which is also true for the labels. In the caption you describe model/port ratio and difference. Which one is true? What is on the y-axis, port or starboard? Please clarify.

Additional references

Bigorre, S. P., Weller, R. A., Edson, J. B., & Ware, J. D. (2013). A surface mooring for air–sea interaction research in the Gulf Stream. Part II: Analysis of the observations and their accuracies. Journal of Atmospheric and Oceanic Technology, 30(3), 450-469.

Emond, M., Vandemark, D., Forsythe, J., Plueddemann, A. J., & Farrar, J. T. (2012). Flow distortion investigation of wind velocity perturbations for two ocean meteorological platforms. Woods Hole Oceanographic Institution.

[Figure]

**List of changes made to amt-2019-366 "Using global reanalysis data to quantify and correct airflow distortion bias in shipborne wind speed measurements"**

- A plot showing the cruise track of the ship for legs 0,1,2,3, and 4 was added as new Figure 1.

- Photographs of the setup of the anemometers on the main mast were added as new Figure 3.

- The former Figure 4 was split in two seperate figures (new Figures 6 and 7).

- The height of the anemometer above the mean water level was re-evaluated the new estimated $31.5\,\mathrm{m}$ a. s. l. is $1\,\mathrm{meter}$ heigher. This has marginal effect on the results.

- The data from the first 10 days of leg 0 was added to the analysis.

- The majority of the observations from leg 2 (277 hours), where removed from the analysis after it became clear that for these data ERA-5 results might be affected by the assimilation of wind speed and direction observations reported by the R/V-*Akademik Tryoshnikov* under the station ID UBXH3.

- The addition and removal of data resulted in slight changes in the estimated flowdistortion bias.

- Appendix section C and Figure C1 was added to contrast results for 5-minute and 1-hour averaging times.

- Appendix section D and Figure D1 was added to Describe the exclusion of data, where ERA-5 results might be affected by the assimilation of wind speed and direction observations reported by the R/V-*Akademik Tryoshnikov* under the station ID UBXH3.

- Appendix section E and Figure E1 was added to compare ERA-5 sea surface temperature and surface air temperature estimates with the in situ observations during legs 1,2, and 3.

[revised manuscript text omitted]

---

## Referee Report (RR1)

**Using global reanalysis data to quantify and correct airflow distortion
bias in shipborne wind speed measurements**

amt-2019-366

Sebastian Landwehr et al.

**General**

The authors did a great job during the revision of the manuscript. I think the changes made regarding the recommendations of the reviewers improved the manuscript. In particular the changes in the figures together with the addition of the photograph are helpful for the reader.

From my point of view all suggestions and recommendations from the first revision were discussed and appropriate changes were made. I recommend publication of the paper after adding one additional reference of a recently published paper regarding flow distortion around buoys.

**Technical comments/suggestions**

Page 3/line 1: Just this week a paper was published (Schlundt et al., 2020), bringing the two references (Emond et al., 2012 and Bigorre et al., 2013) in a wider context by confirming the flow distortion behavior of buoys in the field. Within this paper, the estimated flow distortion results were obtained from long-term records of buoys and related/compared to scatterometer estimates at the particular sites. I think it's worth to cite this recent reference here either.

**Additional reference**

Schlundt, M., J.T. Farrar, S.P. Bigorre, A.J. Plueddemann, and R.A. Weller, 2020: Accuracy of Wind Observations from Open-Ocean Buoys: Correction for Flow Distortion. *J. Atmos. Oceanic Technol.,* **37**, 687–703, https://doi.org/10.1175/JTECH-D-19-0132.1.